# Coherent master equation for laser modelocking

Auro M. Perego[1], Bruno Garbin[2,3], François Gustave[4], Stephane Barland[5], Franco Prati [6] & Germán J. de Valcárcel [7]*

Modelocked lasers constitute the fundamental source of optically-coherent ultrashort-pulsed radiation, with huge impact in science and technology. Their modeling largely rests on the master equation (ME) approach introduced in 1975 by Hermann A. Haus. However, that description fails when the medium dynamics is fast and, ultimately, when light-matter quantum coherence is relevant. Here we set a rigorous and general ME framework, the coherent ME (CME), that overcomes both limitations. The CME predicts strong deviations from Haus ME, which we substantiate through an amplitude-modulated semiconductor laser experiment. Accounting for coherent effects, like the Risken-Nummedal-Graham-Haken multimode instability, we envisage the usefulness of the CME for describing self-modelocking and spontaneous frequency comb formation in quantum-cascade and quantum-dot lasers. Furthermore, the CME paves the way for exploiting the rich phenomenology of coherent effects in laser design, which has been hampered so far by the lack of a coherent ME formalism.

[1] Aston Institute of Photonic Technologies, Aston University, Birmingham B4 7ET, UK. [2] The Dodd–Walls Centre for Photonic and Quantum Technologies, Department of Physics, The University of Auckland, Auckland 1142, New Zealand. [3] Centre de Nanosciences et de Nanotechnologies (C2N), CNRS - Université Paris-Sud - Université Paris-Saclay, Palaiseau, France. [4] DOTA, ONERA, Université Paris-Saclay, F-91123 Palaiseau, France. [5] Université Côte d'Azur, CNRS, INPHYNI, F-06560 Valbonne, France. [6] Dipartimento di Scienza e Alta Tecnologia, Università dell'Insubria, via Valleggio 11, 22100 Como, Italy. [7] Departament d'Òptica i Optometria i Ciències de la Visió, Universitat de València, Dr. Moliner 50, 46100 Burjassot, Spain. *email: german.valcarcel@uv.es

The trains of coherent pulses emitted by modelocked lasers display an impressive list of applications, ranging from telecommunications, sensing and metrology, to health care and materials processing[1,2]. Therefore, laser modelocking is an extremely active research topic with a wealth of open fundamental problems such as a full understanding of its transient dynamics[3], thermodynamical properties[4] and the very modelocking mechanisms[5], including modern proposals allowing spontaneous self-organization and emergence from noise[6–8].

There is in particular a rich literature on the role of so-called coherent effects in laser modelocking, which is the focus of our study. Coherent effects[9] occur when a stable phase relation is established between the light field and the electronic respon of the material. Generally such a phase locking is fragile and coherent effects mostly manifest as transients. In laser systems, however, the long-term interaction between light and matter owed to the resonator feedback allows the persistence of such effects in some cases, e.g., through pulsed operation. We refer to super-fluorescence and other coherent effects in modelocking which give rise to coherent ringing and hyperbolic-secant-shaped pulses shorter than predicted by the standard Siegman-Haus theory[10–14], self-induced transparency modelocking which exploits coherent dynamics of the atomic gain medium and saturable absorber[15–17], Rabi flopping in quantum-cascade lasers[18], spontaneous modelocking via the Risken-Nummedal-Graham-Haken (RNGH) instability[19,20] and superfluorescent and superradiant effects in semiconductor lasers[21], and heterostructures[22], giving rise to superluminal pulse propagation.

So far, all those coherent and cooperative effects in lasers have been modeled using the full set of Maxwell–Bloch equations[5,19,23,24], which hinders analytical treatments and results in heavy simulation processes. In other words, the rich and interesting phenomenology arising from coherent effects in lasers currently lacks a master equation (ME) formalism, and this situation has probably hindered the deployment of such effects in laser design. A coherent ME (CME) theory would allow simple but rigorous description of laser operation in the presence of coherent effects, potentially paving the way to the development of new classes of laser systems that exploit light-matter coherence.

Soon after the pioneering work by Kuizenga and Siegman in the late 1960's[25,26] (also by Haken and Pauthier[27]) Haus' take on the description of laser modelocking via ME[28–31] became the standard approach to the problem. Haus considered first active modelocking via intracavity amplitude- (AM) and phase- (FM) modulation[28], to which followed his treatment on passive modelocking with fast[29] and slow[30] saturable absorbers. Such an approach has proven extremely successful in the description of other modelocking mechanisms like additive-pulse and Kerr-lens modelocking[32,33], related to the combined effect of self-phase modulation and group-velocity dispersion[34].

Despite its popularity, Haus ME approach does not account for coherent effects and, additionally in the case of active modelocking, its validity requires sufficiently slow medium dynamics. In particular, if an extended cavity is required, the pulse period can be comparable with the gain recovery time $T_G$, which is on the order of a nanosecond in many instances, like in semiconductor[35] and dye[36] lasers. A similar problem is encountered in vectorial passive modelocking[37]. The situation is even worse in quantum-dot and quantum-cascade lasers, where $T_G$ is in the picosecond range[38,39]. This circumstance makes passive modelocking with a saturable absorber prohibitive in such lasers and has forced researchers towards alternative solutions, such as active modelocking in an external ring cavity[40]. Yet, even with a cavity length of few millimeters, the pulse period cannot be shorter than $T_G$.

In this paper, we introduce a CME approach for laser modelocking, able to retain light-matter coherence effects for any kind of modelocking mechanism. Further, the CME validity is not affected by the slow-gain limiting condition $T_G \gg T_R$ of Haus ME for active modelocking. In that way we aim at bringing the phenomena associated with fast gain dynamics into the standard framework of the Haus ME, both allowing to obtain analytical insights and providing a more efficient approach to numerical modeling. In order to test the theory we have conducted experiments in a semiconductor laser with an extended cavity. The predictions of our CME differ from Haus ME in several respects, both qualitative and quantitative, and they agree substantially with experiment. Preliminary results of this work have been reported in ref. [41].

## Results

**Limitations of Haus ME approach.** All ME approaches are based on following the changes suffered by a pulse along a full cavity roundtrip, as caused by the different elements like gain, absorber, modulator, dispersive sections, etc. In this way, assuming that the overall change is small, two time scales are introduced, a fast one ($\tau$) describing the pulse shape, and a slow one ($T$) describing pulse evolution along ensuing roundtrips. Then the field complex amplitude $F(T,\tau)$ obeys a differential equation of the type (the ME),

$$T_R \partial_T F = \widehat{\mathcal{R}}[G, \partial_\tau] F, \tag{1}$$

where $T_R$ denotes the cavity roundtrip time and $\widehat{\mathcal{R}}$ is a differential operator that accounts for the changes suffered by the field along one cavity roundtrip, which in particular depends on the gain $G$. Incidentally we note that an alternative, yet related approach to the ME formalism—in the form of delay differential equation model—has been developed which does not require small gain and loss per roundtrip[42].

There are two key points concerning the role of the gain $G$ in ME approaches, namely how $G$ enters $\widehat{\mathcal{R}}$ in Eq. (1) and how its dynamics is modeled. Both are closely related each other, and their analysis determines the weakness of existing ME approaches, as we discuss next.

Regarding the gain dynamics, two main paradigms can be adopted, namely rate equations or Bloch equations. Existing ME models adopt a rate-equation description[43–45] as the gain $G$, being proportional to the population inversion of the lasing transition, has a relaxation rate $T_G^{-1}$ which uses to be several orders of magnitude smaller than the gain bandwidth. In fact two types of rate equations are used in the literature. The most general one can be written as

$$T_G \partial_\tau G = r - G - |F|^2 G, \tag{2}$$

where $r$ is the dimensionless pump parameter. However when the gain takes many cavity roundtrip times to recover ($T_G \gg T_R$) the gain is relatively insensitive to the pulse intensity $|F(T,\tau)|^2$ (the pulse shape), but rather responds to the pulse energy, verifying the equation[28],

$$T_G \frac{dG}{dT} = r - G - PG, \tag{3a}$$

where

$$P(T) = \frac{1}{T_R} \int_{-T_R/2}^{+T_R/2} d\tau |F(T,\tau)|^2 \tag{3b}$$

is the average circulating power (proportional to the pulse energy), which is a slow quantity. Note that (3) is obtained from (2) upon integrating in $\tau$ along one cavity roundtrip, and

neglecting a very small variation of $G$ on the time scale of the pulse duration.

These are excellent approximations in many instances, however, as rate equations ignore the coherent nature of light–matter interaction, they are unable to describe a variety of important effects which substantially modify modelocked pulses features and stability, as we discussed in the Introduction.

Regarding how the effects of the gain are incorporated into Eq. (1), ME approaches do so in ad hoc ways usually following necessity arguments which are specific to each type of modelocking mechanism. Importantly, which of the above rate Eqs. (2) or (3) is used determines how pulse broadening can be modeled. One has to understand that pulse formation and stability requires that the unavoidable pulse broadening brought about by intracavity elements be balanced by some narrowing mechanism introduced into the laser cavity. The latter can be active, e.g., with the use of modulators, or passive, e.g., with the use of saturable absorbers. Clearly this balance can occur only if both mechanisms act on the same time scale, and this scale equality forces the way the gain enters the ME.

Pulse broadening is modeled through a term

$$\widehat{\mathcal{R}}|_{\text{pulse-broadening}} = \Omega_{\text{BW}}^{-2} \partial_\tau^2, \tag{4}$$

in the ME (1), where $\Omega_{\text{BW}}$ is the spectral bandwidth (HWHM) of the bandwidth-limiting element. This form follows from a parabolic approximation to the actual effect in spectral domain, which can be adopted whenever the pulse spectral linewidth is narrow as compared to $\Omega_{\text{BW}}$. The gain element always leads to pulse broadening and in principle it should be the actual bandwith-limiting element: including other dispersive elements decreases the bandwidth, hence increasing pulse duration. However the pulse broadening caused by the amplifier can be modeled as (4) only if the gain does not display relevant variations on the pulse time scale, i.e. if $G$ depends on the slow time scale $T$ but not on $\tau$: only in that case the gain element can be treated as an active filter providing a spectral amplification per roundtrip of the form[25,27,28]

$$\exp \frac{\ell G(T)}{1 + (\omega/\Omega_{\text{G}})^2}, \tag{5}$$

where $\ell$ is the passive linear loss per roundtrip (in our normalization, $G$ represents the gain-to-loss ratio), $\omega$ represents the frequency offset from line center, and $\Omega_{\text{G}}$ is the gain bandwidth of the Lorentzian atomic line. After the commented parabolic approximation (and assuming $\ell \ll 1$) a term like (4) appears, with $\Omega_{\text{BW}}^{-2} = \ell G(T)\Omega_{\text{G}}^{-2}$ (see below for details). Note that this is a quasi-static approximation, valid because one neglects the variations that the gain can display on the pulse duration scale; accordingly the gain is assumed to obey the rate Eq. (3). This type of gain-limited bandwidth ME is found, e.g., in theories of active modelocking[28] and passive modelocking with a fast saturable absorber[29] because in both cases effective pulse-narrowing mechanisms exist which are independent of the gain.

There are other cases, notably passive modelocking with a slow saturable absorber, where stability of pulses requires that the gain itself contributes to the pulse narrowing by dropping below its threshold value after the passage of the pulse[30,46]. This fact forces to consider the response of $G$ to the instantaneous intensity via Eq. (2): $G$ has in this case a $\tau$ dependence and thus the pulse broadening brought about by the gain cannot be modeled by (5) (in fact it cannot be modeled properly in existing ME theories). The solution adopted consists in assuming that an additional

dispersive element (independent of the gain or the saturable absorber sections) is the one that limits the system bandwidth, and in such a case the term (4) is used with given $\Omega_{\text{BW}}$. In words of Haus[30], "Note that the bandwidth-limiting element has introduced the operator $(1/\Omega_{\text{BW}})^2 d^2/dt^2 \dots$ Because diffusion or spreading can be caused by many different physical processes, we surmise that the final equation is relatively model independent." ($\Omega_{\text{BW}}$ is denoted by $\omega_C$ in[30]), or ref. [31] "Here we have expressed the filtering action as produced by a separate fixed filter, rather than by the finite bandwith of the gain (which varies with time) so as to obtain analytic solutions of the master equation."

All the above evidences that existing MEs suffer from fundamental limitations related to the gain dynamics, as well as that each type of modelocking requires a different rationale concerning such dynamics in order to derive a suitable ME, which in some cases can be far from intuitive. Let us introduce next a theoretical framework that overcomes such limitations and allows rigorous and systematic derivation of MEs preserving coherent dynamics.

**Model overview and field map**. We consider a ring-cavity homogeneously broadened two-level laser. The interaction between the traveling-wave light field and the amplifier is modeled with Maxwell–Bloch equations, which constitute the fundamental model at the semiclassical level (see the "Methods" section). In Fabry-Perot cavities a similar treatment can be used whenever no standing-wave patterns exist inside the gain medium, which amounts to place the amplifier section far enough from mirrors so that forward and backward moving pulses do not overlap inside the gain medium. We also note that inclusion of inhomogeneous broadening when necessary does not alter the skeleton of our approach.

Although the approach is valid for any type of modelocking technique, from now on we focus on the case of active modelocking via amplitude modulation. The effect of the modulator is modeled through its transmission function, written as $e^{-m(t)}$, where $m(t)$ represents the modulator state. Our analysis is valid for any periodic function of time $m(t)$, but here we will focus on the classic sinusoidal case

$$m(t) = M[1 - \cos(\Omega_{\text{M}}t)], \tag{6}$$

with $M$ the modulation depth, $\Omega_{\text{M}} \equiv 2\pi/T_{\text{M}}$, and $T_{\text{M}}$ the modulation period. In the case of FM modelocking, $M$ should be substituted by i$M$ simply.

First, we derive the map that describes the overall change suffered by a pulse after one full roundtrip along the cavity, which will be the basis for our CME. Such a procedure requires determining how the amplifier modifies an input, and then following the output along the cavity, through the modulator, till returning to the amplifier input plane.

We assume that the effect on the light field of the amplifier is small. Note that such a hypothesis holds quite generally (though it may not be satisfied in high-power lasers with very large coupling losses): the amplifier action must just balance losses (or other variations) occurring outside the amplifying medium, and they are assumed small. This is valid no matter how long the medium is, nor how close or far from threshold the laser is operating: above lasing threshold saturation acts and gain is always kept small whenever loss is small[47]. Formal integration of the Maxwell Eq. (24a), explicitly taking into account the gain dependence of the group velocity of light in the amplifier, and using the boundary condition imposed by the modulator and the resonator, leads to the sought-for map

(see the "Methods" section),

$$f(0, t + T_{\mathrm{R}}) - f(0, t) = -[\ell + m(t)]f(0, t)$$
$$+ \ell r \left[ p(0, t) + \frac{\overline{D}}{\Omega_{\mathrm{G}}} \partial_t f(0, t) \right]. \qquad (7)$$

Here $f(0, t)$ $(p(0, t))$ is the light field (electric polarization) complex slowly varying amplitude at the medium entrance plane ($z = 0$), $\ell$ is the (very small) loss per roundtrip, $\overline{D}$ is the modulation-period average of the population inversion $D$, $\Omega_{\mathrm{G}}$ is the gain bandwidth, and $r$ is the usual laser pump parameter, which equals 1 at the free-running ($m \to 0$) laser threshold.

**Elimination of the polarization.** The large value of the medium polarization decay rate $\Omega_{\mathrm{G}}$ as compared to that of the population difference $T_{\mathrm{G}}^{-1}$ allows in principle some kind of adiabatic elimination of $p$. The usual adiabatic elimination consists in setting to zero the time derivative of the fast variable, i.e., $\partial_t p \mapsto 0$ in Eq. (24b) in our case. However such a simple elimination leads to $p = Df$, which, although usable in singlemode problems, is clearly inadequate in multimode scenarios because it overlooks pulse broadening. Instead we solve formally Eq. (24b) as ref. [48–52]

$$p = \left(1 + \Omega_{\mathrm{G}}^{-1}\partial_t\right)^{-1} Df. \qquad (8)$$

In Fourier domain the differential operator reads as $(1 - i\omega/\Omega_{\mathrm{G}})^{-1}$, and we adopt its parabolic approximation around $\omega = 0$, namely $\left(1 + i\omega/\Omega_{\mathrm{G}} - (\omega/\Omega_{\mathrm{G}})^2\right)$, valid when the pulse linewidth is much narrower than the gain bandwidth. This translates into the time domain as

$$p = \widehat{\mathcal{L}}_t Df, \qquad (9a)$$

where

$$\widehat{\mathcal{L}}_t \equiv 1 - \Omega_{\mathrm{G}}^{-1}\partial_t + \Omega_{\mathrm{G}}^{-2}\partial_t^2 . \qquad (9b)$$

Expression (9b) represents the minimal expansion accounting for the finite spectral bandwidth of the gain. In case of ultrashort pulses whose linewidth is comparable with the gain bandwidth one can extend the expansion or even keep the full differential operator as it is in Eq. (8). In particular, the latter would open the way to describe ultrashort pulses caused by coherent broadening[13]. We point out that the approximation (9) has been successfully used in particular to tackle the self modelocking that emerges via the RNGH instability, even in the presence of large losses[52].

Note that if the inversion fast dynamics is neglected as in Haus ME for active modelocking, then $D \to \overline{D}$, Eq. (9a) becomes $p = \overline{D}\widehat{\mathcal{L}}_t f$, and the last bracketed terms in Eq. (7) turn into

$$\overline{D}f(0, t) + \frac{\overline{D}}{\Omega_{\mathrm{G}}^2} \frac{\partial^2}{\partial t^2} f(0, t) ,$$

which match the gain-dependent terms in Haus ME[28,31] (more on this below).

From now on we will use the generalized adiabatic expression (9) for the polarization.

**Transforming the map into a ME via introduction of two times.** According to the field map (7) and to expression (9a) we have been able to describe the laser dynamics in terms of variables $f(0, t)$, $D(0, t)$, i.e., the field and the inversion at the amplifier entrance plane $z = 0$. We also observe that the inversion $D$ appears multiplied by the pump parameter $r$ (37) in the map, both in $\overline{D}$ and through $p$, which is linear in $D$. Accordingly we introduce the following notation to keep expressions simpler and

also to pave the way to a well-founded definition of the ME:

$$\mathcal{F}_n(t') \equiv f(0, nT_{\mathrm{R}} + t'), \qquad (10a)$$

$$\mathcal{G}_n(t') \equiv rD(0, nT_{\mathrm{R}} + t'), \qquad (10b)$$

$n$ integer, where in particular we have defined the function $\mathcal{G}_n(t')$, which we will refer to as the gain. We apply this notation to the map (7) and obtain

$$\mathcal{F}_{n+1}(t') - \mathcal{F}_n(t') = -[\ell + m_n(t')]\mathcal{F}_n(t') + \ell\widehat{\mathcal{L}}_{t'}\mathcal{G}_n(t')\mathcal{F}_n(t')$$
$$+ \ell(\overline{\mathcal{G}}_n/\Omega_{\mathrm{G}})\partial_{t'}\mathcal{F}_n(t'), \qquad (11)$$

where $m_n(t') = m(nT_{\mathrm{R}} + t')$. Recalling (6) and subtracting $n\Omega_{\mathrm{M}}T_{\mathrm{M}} = 2n\pi$ from the argument of the cosine, the modulation function can be rewritten as $m_n(t') = M\{1 - \cos[\Omega_{\mathrm{M}}(t' - \theta nT_{\mathrm{R}})]\}$, being $\theta = (T_{\mathrm{M}} - T_{\mathrm{R}})/T_{\mathrm{R}}$ the modulation-period mismatch per roundtrip. We have removed $2n\pi$ from the argument of the cosine for mathematical convenience (the transformation is exact): we wish that the modulation function varies slowly on the quantity $nT_{\mathrm{R}}$ (we anticipate that $\theta$ will be very small).

Next, in the spirit of the original Haus ME[28], we introduce a slow time $T'$ that counts the number of roundtrips. We also introduce continuous fields $F'(T', t')$ and $G'(T', t')$ via

$$F'(T' = nT_{\mathrm{R}}, t') = \mathcal{F}_n(t'), \qquad (12a)$$

$$G'(T' = nT_{\mathrm{R}}, t') = \mathcal{G}_n(t'), \qquad (12b)$$

so that the left-hand side of map (11) is approximated by $T_{\mathrm{R}}\partial_{T'}F'(T', t')$, and the laser equations become

$$\frac{T_{\mathrm{R}}}{\ell} \frac{\partial F'}{\partial T'} = -\left(1 + \frac{m'}{\ell}\right)F' + \widehat{\mathcal{L}}_{t'}G'F' + \frac{\overline{G'}}{\Omega_{\mathrm{G}}} \frac{\partial F'}{\partial t'} , \qquad (13a)$$

$$T_{\mathrm{G}} \frac{\partial G'}{\partial t'} = r - G' - \mathrm{Re}\left(F'^* \widehat{\mathcal{L}}_{t'}G'F'\right) , \qquad (13b)$$

where $m'(T', t') = M\{1 - \cos[\Omega_{\mathrm{M}}(t' - \theta T')]\}$, and Eq. (13b) derives straightforwardly from Eq. (24c) after multiplying it by $r$, and setting $z = 0$. We remind that $\overline{G'}$ is the average of $G'$ in one period.

There are two features in these equations that make difficult their analysis and numerical simulation:

(1) Equations (13) verify the following asynchronous boundary conditions

$$X'(T' + T_{\mathrm{R}}, t') = X'(T', t' + T_{\mathrm{R}}), X \in \{F, G\}, \qquad (14)$$

by construction, and

(2) The modulation function $m'(T', t')$ moves at a speed equal to $\theta$, because the map (11) it derives from is stroboscopic with period $T_{\mathrm{R}}$, which differs in general from the modulation period $T_{\mathrm{M}}$.

We remove both drawbacks by introducing the times

$$T = T' + t', \tau = t' - \theta T', \qquad (15a)$$

and fields

$$X(T, \tau) = X'(T', t'), X \in \{F, G\}, \qquad (15b)$$

together with the chain rule for differentiation

$$\partial_{T'} \to \partial_T - \theta\partial_\tau, \ \partial_{t'} \to \partial_T + \partial_\tau. \qquad (15c)$$

This way the fields obey, by construction, standard isochronous (in $T$) periodic boundary conditions (in $\tau$),

$$X(T, \tau) = X(T, \tau + T_{\mathrm{M}}), X \in \{F, G\}, \qquad (16)$$

hence we restrict the problem to $\tau \in [-T_M/2, T_M/2]$ for definiteness. Finally the modulation function becomes

$$m(\tau) = M[1 - \cos(\Omega_M \tau)], \tag{17}$$

recovering its original expression (6). As is customary we will replace $m(\tau)$ with its lowest-order Taylor expansion around $\tau = 0$[28,31], namely $\frac{1}{2} M \Omega_M^2 \tau^2$, as pulses can exist only close enough to the minimal-loss point, given by $m = 0$.

**The coherent master equation**. In order to make contact with Haus ME we split the gain in its modulation-period average

$$\overline{G}(T) = \frac{1}{T_M} \int_{-T_M/2}^{T_M/2} G(T, \tau) \mathrm{d}\tau , \tag{18a}$$

and its remainder

$$g(T, \tau) = G(T, \tau) - \overline{G}(T) , \tag{18b}$$

which is neglected in Haus treatment of active modelocking.

The dynamical equations of these variables derive from using Eqs. (15) in (13b) and averaging:

$$T_G \partial_T \overline{G} = \overline{\Gamma}(T), \tag{19a}$$

$$T_G(\partial_T + \partial_\tau)g = \Gamma(T, \tau) - \overline{\Gamma}(T), \tag{19b}$$

where $\Gamma(T, \tau)$ denotes the right-hand side of Eq. (13b) after using (15). As $g$ is a fast variable, $\partial_T g \ll \partial_\tau g$, hence we approximate Eq. (19b) with

$$T_G \partial_\tau g = \Gamma(T, \tau) - \overline{\Gamma}(T), \tag{19c}$$

which is a kind of adiabatic elimination of the fast gain. This rationale must be further applied for consistency to every equation. In particular, in the right-hand side of Eq. (13a) transformed by (15) we set $\partial_{t'} = \partial_\tau$ and $\hat{\mathcal{L}}_{t'} = \hat{\mathcal{L}}_\tau$, because $F$ and $g$ are fast variables; i.e., when applying the rule (15c) to Eq. (13a) we neglect the derivative $\partial_T$ as before. Note that such an approximation cannot be taken in the derivative $\partial_{T'}$ of the left-hand side because $\theta$ can be zero. The same approximations must be done in the right-hand sides of Eqs. (19a) and (19c), i.e., in the function $\Gamma(T, \tau)$.

The equations obtained as explained above constitute what we call CME; however we find numerically that the gain equations contain very small terms that can be dropped with negligible influence in the final result. Thus we adopt a final approximation, which merely speeds up the simulations: In the function $\Gamma(T, \tau)$,

(i)　$\hat{\mathcal{L}}_\tau \to 1 - \Omega_G^{-1}\partial_\tau$, i.e. we drop all second-order derivatives on $\tau$, and

(ii)　further we neglect terms containing $\partial_\tau(gF)$.

This way we end up with the CME, which reads

$$\frac{T_R}{\ell} \frac{\partial F}{\partial T} = \left( \overline{G} - 1 - \mu^2 \Omega_M^2 \tau^2 + \tau_d \partial_\tau + \overline{G} \frac{\partial_\tau^2}{\Omega_G^2} \right) F + \left( 1 - \frac{\partial_\tau}{\Omega_G} + \frac{\partial_\tau^2}{\Omega_G^2} \right) (gF), \tag{20a}$$

$$T_G \frac{\mathrm{d}\overline{G}}{\mathrm{d}T} = r - \overline{G}\left(1 + \overline{|F|^2}\right) - \overline{g|F|^2} , \tag{20b}$$

$$T_G \partial_\tau g = -\left(1 + |F|^2\right)g + \overline{G}\left(\overline{|F|^2} - |F|^2\right) + \frac{\overline{G}}{2} \frac{\partial_\tau}{\Omega_G} |F|^2 + \overline{g|F|^2}, \tag{20c}$$

where

$$\tau_d = (T_M - T_R)/\ell , \tag{21a}$$

is the modulation detuning, and

$$\mu^2 = M/2\ell . \tag{21b}$$

We note that the validity of the final approximations (i) and (ii) above has been assessed numerically under diverse sets of parameters ($\Omega_G$ from $10^{12}\,\mathrm{s}^{-1}$ to $10^{13}\,\mathrm{s}^{-1}$, $T_G$ from 0.5 ns to 1 µs, and $T_R \lesssim T_G$), which cover usual solid-state and semiconductor lasers. We observe that the approximations start to break down for much longer cavities, a situation that favors the size increase of the fast gain component $g$. In any case, if in a specific application approximations (i) and (ii) did not hold, one would end up with a CME like (20) but with some extra terms, which however would not change the mathematical structure of the equations. Regarding such a structure, some comments are in order:

(a)　The two times play different roles: $T \in [0, +\infty)$ and is the time-like coordinate on which the system evolves, while $\tau \in [-T_M/2, +T_M/2]$ and is the space-like coordinate on which $F$ and $g$ verify the periodic boundary conditions (16) ($\overline{G}$ is a mean field which only depends on $T$).

(b)　$F$ and $\overline{G}$ obey the dynamical Eq. (20a) and (20b), while $g$ follows adiabatically the other quantities: Eq. (20c) is not an evolution equation as it allows determining, at every instant $T$, the value of $g(T, \tau)$ in terms of $F(T, \tau)$ and $\overline{G}(T)$. This means that, despite appearances, the complexity of the CME is mathematically similar to that of Haus ME, which we recover below.

**The CME and the RNGH instability**. A clear evidence that the CME (20) preserves coherent effects is its ability to capture the laser multimode RNGH instability. This instability affects the singlemode lasing solution above a certain pump level, leading to spontaneous self modelocking, and is an acid test for coherent laser models.

Predicted back in 1968[53,54] the RNGH instability is due to the sideband gain originating from coherent Rabi pulsations[55], and was considered as an academic curiosity for almost 30 years. However the scenario changed dramatically in 1997 when Pessina and coworkers reported experimental observations that could be a manifestation of the RNGH instability in an erbium-doped fiber laser[56]. That work triggered theoretical and experimental research that allowed gaining insight into the role of the RNGH instability in fiber lasers[57] as well as in general terms[58,59] (see ref. [60] for a review till year 2005). More recently the RNGH instability has gained popularity as it has been invoked in order to understand the hot topics of spontaneous frequency-comb formation and self modelocking in quantum-dot and quantum-cascade lasers[19,20,24,61,62].

The CME (20) with $\mu = \tau_d = 0$ describes a free-running laser and accounts for the usual singlemode lasing solution. A linear stability analysis of such a solution (see the "Methods" section) reveals that coupled sidebands, symmetrically detuned by $\pm \omega$ from the lasing mode, experience net gain above a threshold, signaling a multimode instability. The phenomenon is described by a complex eigenvalue of the linear problem, $\lambda_X$, whose real part governs the instability growth. In the relevant limit $\Omega_G T_G \gg 1$, we get

$$\mathrm{Re}\lambda_X = (\Omega_G T_G)^{-1}\left[3(r-1) - \tilde{\omega}^2 - \frac{2r(r-1)}{\tilde{\omega}^2}\right], \tag{22}$$

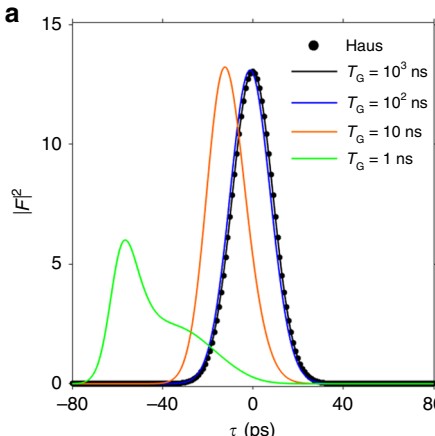
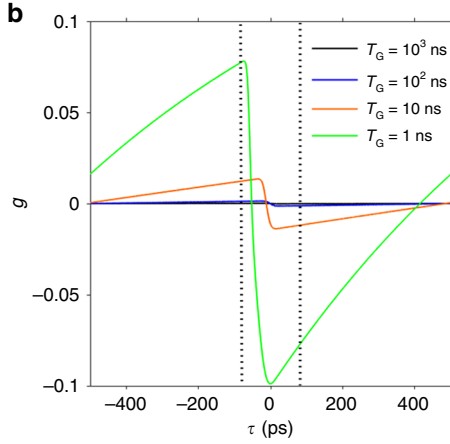

**Fig. 1 Dependence of modelocked pulses on the gain recovery time.** In (**a**) the pulses calculated from CME (20) for four different values of the gain recovery time $T_G$ (decreasing from right to left) are shown and compared with that of Haus ME (23). As $T_G$ decreases the pulse shapes departs more and more from the Gaussian centered at $\tau = 0$. In (**b**) the corresponding profile of the fast gain is shown: $g$ increases in magnitude for faster gain media. The vertical dashed black lines denote the time interval shown in (**a**). Parameters used are $\Omega_G = 10^{12}$ s$^{-1}$, $T_M = T_R = 1$ ns, $\ell = 0.6$, $M = 1.2$, $r = 1.3$.

to the leading order, with $\tilde{\omega} \equiv \omega\sqrt{T_G/\Omega_G}$ the normalized sideband frequency. This expression matches the prediction of the full Maxwell–Bloch equations[52–54,58,60] in the considered limit, which proves that the CME (20) contains the RNGH coherent instability. On the contrary, Haus ME does not account for this result.

**Recovering Haus ME**. If, in the spirit of Haus approach, we neglect the fast component of the gain by arbitrarily setting $g = 0$ in Eqs. (20), we obtain

$$\frac{T_R}{\ell}\frac{\partial F}{\partial T} = \left(\overline{G} - 1 - \mu^2\Omega_M^2\tau^2 + \tau_d\partial_\tau + \overline{G}\frac{\partial_\tau^2}{\Omega_G^2}\right)F , \quad (23a)$$

$$T_G\frac{d\overline{G}}{dT} = r - \overline{G}\left(1 + \overline{|F|^2}\right) , \quad (23b)$$

which is the celebrated Haus ME for AM modelocking[28,31,35]. Equations (23) hold the symmetry $(\tau_d, \tau) \longmapsto (-\tau_d, -\tau)$, meaning that the only role played by the sign of the asynchrony $\tau_d$ is to reverse the fast time $\tau$, having no consequence on the system dynamics. As it is well known, in the steady state ($\partial_T \to 0$) the stable solutions of Eq. (23) are Gaussian pulses[25,28] (see the "Methods" section).

**The CME vs. Haus ME**. The CME (20) has been integrated numerically using two different methods (see the "Methods" section). First, we compare the predictions of the CME vs Haus ME.

In Fig. 1 we show the results obtained for different values of $T_G$ and fixed $T_R$. In the limit $T_G \gg T_R$, which is typical of solid-state, fiber or gas lasers (provided that the cavity is not extremely long) the gain saturation can be considered constant across the pulse profile and the classic Gaussian pulse predicted by Haus ME is found. However, if $T_G \gtrsim T_R$ a very different solution is obtained (Fig. 1a) which is accompanied by an increase of $g$ (Fig. 1b). Thus we can ascribe the qualitative change in the pulse shape when $T_G \gtrsim T_R$ to the fast gain component $g$ arising from the coherent light-matter interaction, neglected in the original approach by Haus. As well, according to Haus ME, if $T_M = T_R$ the pulse is centered at $\tau = 0$, while this no longer holds in our CME when $T_G$ approaches $T_R$ (Fig. 1a). The fast gain component (which is positive to the left of the pulse peak) exerts a "force" on the pulse pulling it towards negative temporal coordinates. By applying a certain modulation frequency detuning $\Delta \equiv T_M^{-1} - T_R^{-1} (\approx -\ell\tau_d/T_R^2)$ the pulse can be brought

back to $\tau = 0$. Correspondingly, the pulse intensity is maximum although it still preserves a slight asymmetry in its tails.

**The experiment**. The experimental setup is sketched in Fig. 2, and consists of an amplitude-modulated ring laser based on a semiconductor amplifier (see the "Methods" section).

The optical path length is about 9 m, which corresponds to a cavity roundtrip time $T_{R,exp} \approx 29$ ns, and a field lifetime of tens of nanoseconds. The choice of such a long cavity was motivated by the will of exploring the limit $T_R \gg T_G$ where Haus theory was expected to fail. Due to the very large roundtrip time the laser losses can be modulated around different harmonics of the cavity free spectral range, which is convenient for varying in a simple way the effective cavity roundtrip time. Although the CME (20) was derived for fundamental modelocking we can still apply it to the analysis of these experimental results considering that the various pulses that coexist in the cavity do not interact because of the strong modulation of losses. Hence we regard them as belonging to $n$ virtually independent lasers[63,64], if $n$ is the harmonic order of the modulation. Therefore, in the simulations of Eq. (20) we simply set $T_R = T_{R,exp}/n$.

Extensive experiments have been carried out from harmonic $n = 14$ (modulation frequency $T_M^{-1} \approx 484$ MHz) up to $n = 175$ ($T_M^{-1} \approx 5995$ MHz). Faster and faster dynamics take place at higher harmonic number and the limited detection bandwidth gradually filters out most of the signal. In addition, lower harmonic numbers display slower dynamics and smaller locking range due to the long positive net gain window opening. Here we choose to present results obtained around harmonic $n = 25$ ($T_M^{-1} \approx 866$ MHz), leading to the simultaneous circulation of 25 equidistant pulses along the cavity.

Deeply in the locked regime distinct pulses show no appreciable differences. On the contrary, at the edges of the locking region both the detailed spatio-temporal plot and the average trace of the pulse over successive roundtrips often differ between statistical samples, which confirms the coherence of one pulse from one roundtrip to the next but the loss of memory between two consecutive optical pulses (separated by about 1.15 ns) within one cavity roundtrip. See Methods for details about the data acquisition and processing.

**Analysis of experimental results**. We have compared the predictions of our CME (20) with experimental results obtained in

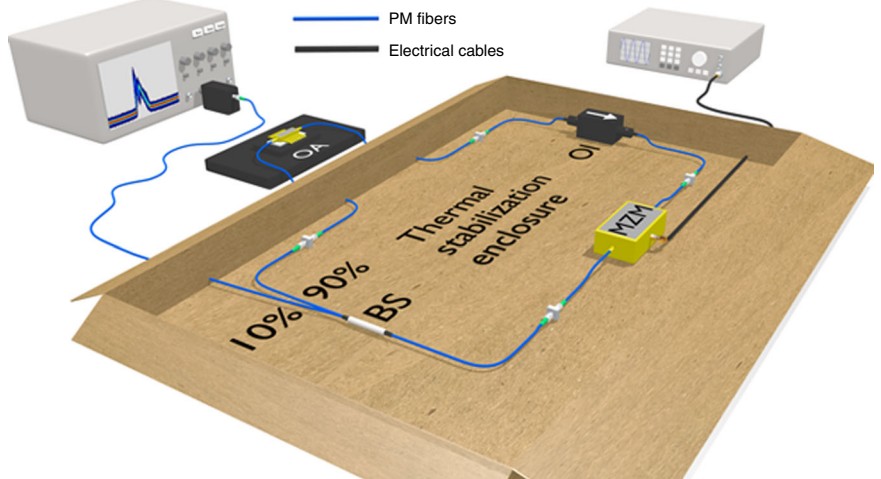

**Fig. 2 Experimental setup.** The actively modelocked laser is based on an antireflection coated semiconductor optical amplifier operated in a closed loop including an optical diode for unidirectional operation, an intensity modulator and a low transmission fiber beam splitter allowing signal detection. OA fiber-coupled semiconductor amplifier, PM single mode polarization-maintaining fiber, OI optical isolator, MZM Mach-Zehnder lithium niobate intensity modulator.

the ring-cavity semiconductor laser, for different effective values of the cavity roundtrip time $T_R$. Here we show results for $T_R = 1.155$ ns, which is of the same order of magnitude as the gain recovery time of semiconductors (we use $T_G = 0.75$ ns in the following numerical simulations), a situation where strong deviations from Haus theory are expected.

The long-term intensity of the pulses is plotted versus modulator detuning in Fig. 3, corresponding to our CME (20) (Fig. 3a), the experiment (Fig. 3b), and Haus ME (Fig. 3c).

The three most typical pulse shapes are shown in Fig. 4 for the CME (20) (Figs. 4a, c) and for the experiment (Figs. 4d, f).

Finally the most symmetric pulses are analyzed in Fig. 5, corresponding to our CME (Fig. 5a) and the experiment (Fig. 5b).

The detuning interval over which stable pulses can be found is similar in CME simulations and in the experiment. In the experiment, outside the locked regime, the width of the peaks in the radiofrequency spectrum indicate that the mode spacing of the device is not defined to a better precision than a few tens of kHz. Therefore, the detuning condition cannot be precisely determined, and the vertical axis in Fig. 3b is in absolute units. Also note that the origin of time $t = 0$ is arbitrary due to the presence of numerous electronic and optical delays between the modulation and the laser signals.

The asymmetric action of the detuning predicted by the CME (Fig. 3a) and already observed in some earlier experiment[65] is experimentally confirmed (Fig. 3b). Haus ME fails in this regard because it predicts that stable pulses are always Gaussian and there is no difference between positive and negative detuning $\tau_d$ besides a trivial mirror-symmetric dynamics[66] (see the "Methods" section).

In particular, a main effect of the detuning in Haus ME is a shift of the Gaussian pulse with respect to the minimum loss point $\tau = 0$ proportional to $\tau_d$. In our CME, instead, the pulse shape depends strongly on the sign of the detuning. Mathematically, this asymmetry follows from the term $\partial_\tau(gF)$ in Eq. (20a) and from the very structure of Eq. (20c), which prevents the invariance of the full CME set (20) under the simultaneous changes $\tau_d \to -\tau_d$, and $\tau \to -\tau$. This agrees with the findings of ref. [67], where a similar shift was observed in the framework of a model for the propagation of fast pulses in a fiber which goes beyond the rate equation approximation, although it neglects gain coherent dynamics.

Our theory also captures well the changes in the shape of the pulses that are observed in the experiment as the modulation period $T_M$ is reduced. As shown in Fig. 4 the typical sequence by increasing the modulation frequency is: asymmetric bell-shaped pulses, pulses with a bump in the right tail, intense symmetric pulses.

A quantitative discrepancy can be noticed concerning the pulse duration, which is typically longer in the experiment than in the theory except in the broader and weaker pulses of Figs. 4a, d. Such a discrepancy was already observed in the earlier experiments on active modelocking, for instance in a Nd:YAG laser[26] and in a dye laser[36], and was ascribed to the etalon effect caused by the various intracavity elements. The theoretical treatment remains valid but the effective bandwidth of the atomic gain curve is drastically reduced and the pulsewidth is much wider than expected[63]. Specifically, we have determined that parasitic reflections at the amplitude modulator are responsible for the reduced spectral width. In addition, our model for a two-level system cannot fully capture the whole physics of a semiconductor amplifier, in particular by neglecting the linewidth enhancement factor (Henry's $\alpha$ factor) of semiconductors, and this could have an impact on some important features of the pulses.

Another relevant feature of the experimental pulses, even of the most symmetric ones, is their non-Gaussianity as shown in Fig. 5, a fact accounted for by CME (20) but clearly not by Haus ME. Actually the experimental pulses are sech type, a fact already pointed out in early experimental studies of coherent effects of modelocking in argon lasers (compare our Fig. 5 with Fig. 2 of ref. [12]). Our CME predicts pulses that interpolate between Gaussians and hyperbolic secants, and a clue of why this is happening comes from observing the main contribution to the pulse shape of the fast gain component $g$ in Eq. (20a), which is $-\Omega_G^{-1}(\partial_\tau g)F$. In its turn the main contribution to the derivative comes from the pulse intensity in Eq. (20c), namely $\partial_\tau g \approx (\overline{G}/T_G)\left(\overline{|F|^2} - |F|^2\right)$. Hence there is a term $+(\overline{G}/\Omega_G T_G)|F|^2 F$ in the field equation which has the same form as in the ME for passive modelocking via fast saturable absorber[29], whose solutions are hyperbolic secants[31,70]. Certainly there are additional terms in the CME (20) contributing to the pulse shape, notably the active modulation term proportional to $\tau^2$, which is responsible for the Gaussianity. Apparently in the experiment the

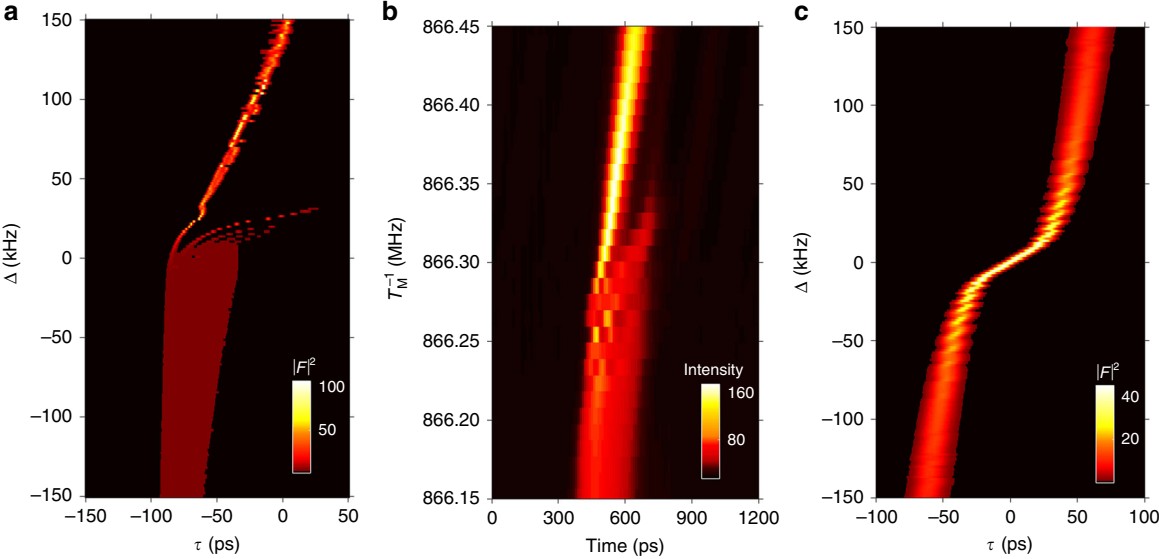

**Fig. 3 Dependence of modelocking on the modulation frequency.** The long-term intensity of pulses averaged over few modulation periods is displayed. Numerical results from the CME (Eq. (20)) are shown in (**a**) for different values of the modulator detuning $\Delta = T_M^{-1} - T_R^{-1}$, and compared to the experimental results in (**b**) for different values of the modulation frequency $T_M^{-1}$, showing good agreement. For reference, the results obtained from Haus theory are shown in (**c**). The CME captures an essential feature of modelocking observed in experiment, namely the asymmetric effect of the detuning which is absent in Haus theory. The parameters used in (**a**) and (**c**) are $\Omega_G = 10^{13}$ s$^{-1}$, $T_R = 1.155$ ns, $\ell = 0.6$, $M = 1.2$, $r = 1.3$ and $T_G = 0.75$ ns.

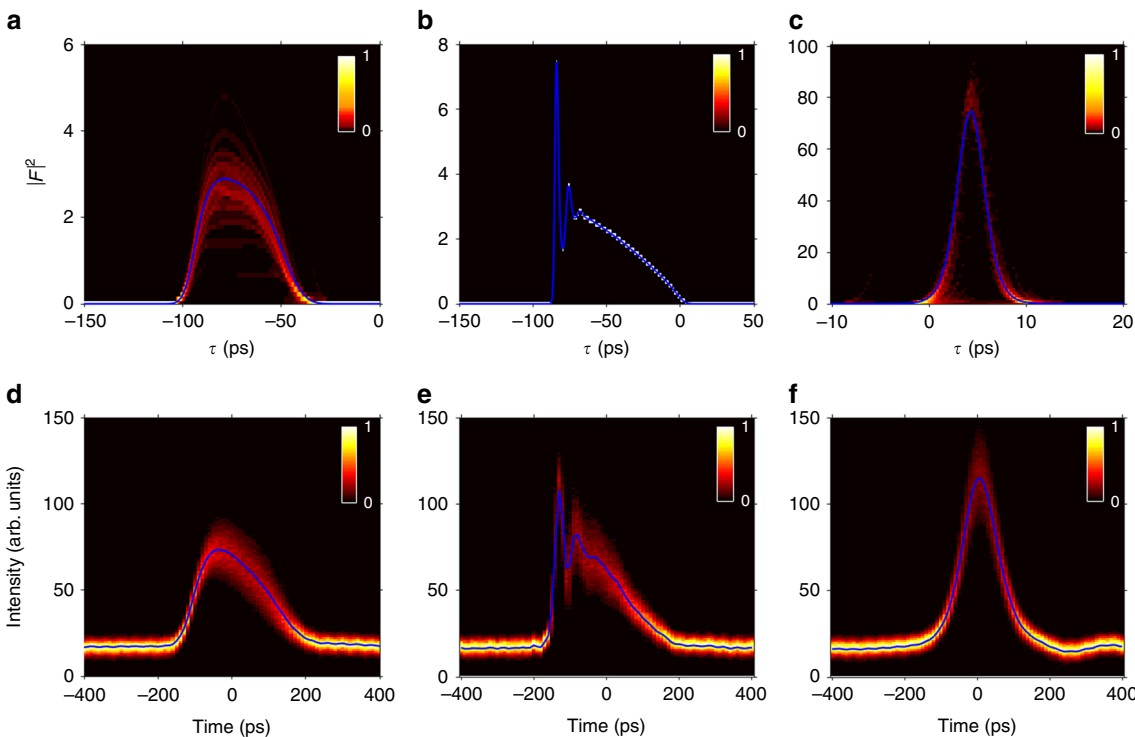

**Fig. 4 Typical shapes of pulses.** The three paradigmatic pulse shapes predicted theoretically (**a**), (**b**) and (**c**) are compared with their experimentally observed counterparts: (**d**), (**e**) and (**f**) (the experimental plots show the pulse intensity statistics, light and dark colors correspond to more and less frequent events respectively, while the blue lines show the average pulse intensity). The vertical axis in experimental plots refer to the photocurrent measured by the detector. Same parameters as in Fig. 3, with $\Delta = -101.73$, 2.63 and 148.15 kHz in (**a**), (**b**), and (**c**) respectively.

role of the modulation on the pulse shape is less pronounced in certain cases, and this should be further considered in the future. In any case we can conclude that the CME (20) can explain, at least partially, why the sech profile was observed in early experiments[12], while our experimental observations indicate that such a qualitative change is robust.

## Discussion

We have set a framework for the study of laser modelocking via CME. The approach put forward here is systematic and preserves the atomic coherence due to light-matter interaction in such a consistent and accurate way that is able to account for the self-modelocking initiated by the RNGH coherent instability,

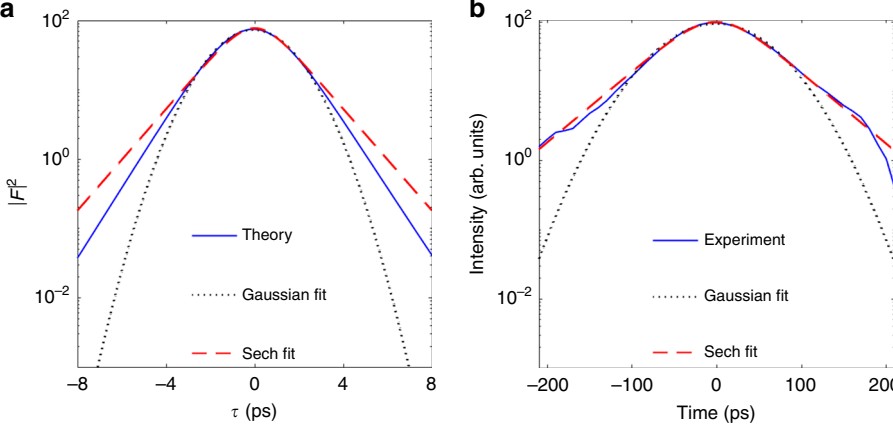

**Fig. 5 Shape of the most symmetric pulses.** The intensity of the most symmetric theoretical (**a**) and experimental (**b**) pulse is depicted in logarithmic scale together with Gaussian and hyperbolic-secant fits. It is evident that pulse tails deviate substantially from the Gaussian shape predicted by Haus theory, exhibiting a smoother decay. The pulses shown here correspond respectively to those in Fig. 4c, f.

responsible for quantum-cascade laser spontaneous modelocking and frequency comb generation.

The structure of the CME (20) is mathematically similar to Haus ME (23) in that there are two dynamical variables, namely the field $F$ and the (slow) gain $\overline{G}$, each ruled by a first-order differential equation on the slow time $T$, whereas the newly introduced fast component of the gain $g$ depends instantaneously (in $T$) on the true dynamical variables. The field $F$ (also the gain $g$) rigorously verifies periodic boundary conditions (16) by construction, due to a proper definition of the slow and fast time scales.

In the limit of slow gain, defined by $T_R/T_G \ll 1$, our CME and Haus ME yield virtually identical results, as expected. However, as the ratio $T_R/T_G \rightarrow 1$ the situation changes dramatically: The divergent predictions of our CME with respect to Haus ME in that limit have been experimentally confirmed, evidencing that coherence in light-matter interaction, an often neglected feature in applied laser physics, and fast gain dynamics can play a significant role in laser modelocking, heavily affecting the shape and the dynamics of pulses. In particular we have been able to trace the experimentally observed hyperbolic-secant-like shape of pulses under active modelocking to an effective fast saturable absorption response present in the CME, owed to the amplifier itself.

The CME can be generalized to a variety of active (e.g., pump-current modulation) and passive (e.g., saturable absorber, Kerr lens) modelocking techniques. In particular, we observe that the very large ratio $T_R/T_G$ needed for the emergence of dissipative solitons in the context of passive modelocking[68] should require using our CME properly adapted to the setup in order to reveal possible coherent effects. Further, our theory could generalize the cubic-quintic complex Ginzburg-Landau equation describing dissipative solitons[69,70], to the case in which gain dynamics cannot be neglected. Moreover, the CME can account for the most diverse intracavity effects, including spontaneous modelocking initiated by the RNGH instability, superfluorescent and superradiant coherent effects.

All this makes the CME a flexible, numerically efficient, compact, and accurate fundamental platform for laser modelocking description, which solves a longstanding and overlooked fundamental problem. The CME is particularly suitable for semiconductor lasers, including quantum dots and especially quantum-cascade lasers with potentially disruptive applications for the generation of frequency combs in the mid infrared region of the electromagnetic spectrum. From a general perspective we expect that the CME will contribute to the development of new types of lasers, based on the rich phenomenology associated to the coherent effects of light-matter interaction.

## Methods

**Maxwell–Bloch equations and boundary condition.** The standard Maxwell–Bloch equations for a homogeneously broadened two-level medium can be written as[43,45],

$$\partial_z f + v_0^{-1}\partial_t f = (a/2)p , \quad (24a)$$

$$\partial_t p = \Omega_G(Df - p) , \quad (24b)$$

$$T_G\partial_t D = 1 - D - \mathrm{Re}(p^*f) , \quad (24c)$$

where $z$ is the light propagation direction and $t$ is time. The actual light electric field, electric polarization, and population inversion are respectively proportional to $\mathrm{Re}[f(z,t)e^{i(k_0z-\omega_0t)}]$, $\mathrm{Im}[p(z,t)e^{i(k_0z-\omega_0t)}]$, and $D(z,t)$, where $\omega_0$ is the atomic transition frequency and $k_0 = n_0\omega_0/c$, being $n_0$ the background refractive index at the frequency $\omega_0$. As we are considering the relevant case when typically thousands of cavity modes fall under the gain line, we assume that one of such modes is in resonance with the atomic transition, and then $\omega_0$ is interpreted also as the resonant cavity mode frequency.

All three variables $(f, p, D)$ are dimensionless and, in particular $|f|^2$ represents the light intensity normalized to its saturation value. As for the parameters, $v_0$ is the background group velocity (without taking into account the effect of the two-level transition), $a$ is the unsaturated gain per unit length (proportional to the pump), $\Omega_G$ is the HWHM gain bandwidth, and $T_G$ is the population inversion (gain) lifetime. Typical values for the time constants of semiconductor, solid-state, and fiber lasers actually span several orders of magnitude: $\Omega_G^{-1} \sim 100$ fs – 1 ps, and $T_G \sim 1$ ps – 100 μs. Note that in the modern notation of laser dynamics studies, $\Omega_G$ and $T_G^{-1}$ are usually denoted by $\gamma_\perp$, resp. $\gamma_\parallel$[43,45].

Equations (24) are supplemented by the boundary condition for the light field as imposed by the cavity and the intracavity modulator: Respectively denoting by $z = 0$ and by $z = w$ the entrance and exit planes of the active medium, and assuming a ring cavity for simplicity, the boundary condition reads

$$f(0, t) = e^{-[\ell+m(t)]}f(w, t - t_e). \quad (25)$$

Here $\ell$ is the linear loss coefficient of the resonator (including the minimum loss introduced by the modulator), while $m(t)$ describes the modulator transmission state.

The delay $t_e$ in Eq. (25) is the travel time of light along the cavity, from the medium exit plane back to the medium entrance plane, so that

$$T_R^{\mathrm{cold}} \equiv t_e + w/v_0, \quad (26)$$

is the cold-cavity roundtrip time. The amplifier modifies the group velocity of light from its cold-cavity value $v_0$ to a value $v$, so that the actual cavity roundtrip time reads

$$T_R \equiv T_R^{\mathrm{cold}} + \delta T_R, \quad (27)$$

with

$$\delta T_R \equiv w(v^{-1} - v_0^{-1}), \tag{28}$$

an increment which we evaluate below.

Next we shift time by $T_R$ in Eq. (25) for convenience,

$$f(0, t + T_R) = e^{-[\ell + m(t)]} f(w, t + w/v), \tag{29}$$

where we wrote $m(t)$ instead of $m(t + T_R)$ (it is a redefinition), and used the relation $T_R - t_e = w/v$, as follows from Eqs. (26), (27), and (28). As losses are assumed small, as is the effect of the modulator, the exponential in (30) is customarily Taylor expanded[28] as $1 - \ell - m(t)$,

$$f(0, t + T_R) = [1 - \ell - m(t)] f(w, t + w/v), \tag{30}$$

which is the form of the boundary condition we will use.

Let us now evaluate (28), which requires determining the group velocity $v$. We do so from the dispersion relation of the medium as usual: we consider the propagation of a monochromatic wave $f(z, t) = f_\omega(z)e^{-i\omega t}$, detuned by $\omega$ from the atomic resonance. The polarization has a similar expression, $p(z, t) = p_\omega(z)e^{-i\omega t}$, while the population inversion will be a constant, $D$. Substitution of these expressions into Eqs. (24a) and (24b) leads to the result $f_\omega(z) = f_\omega(0)e^{ikz}$, where

$$k(\omega) = \left( \frac{\omega}{v_0} + \frac{aD}{2} \frac{\Omega_G \omega}{\Omega_G^2 + \omega^2} \right) - i\frac{aD}{2} \frac{\Omega_G^2}{\Omega_G^2 + \omega^2} . \tag{31}$$

The inverse group velocity is then $v^{-1} = \text{Re}(dk/d\omega)_{\omega=0}$ ($\text{Im}(k)$ accounts for amplification) which substituted into Eq. (28) yields

$$\delta T_R = \frac{aw}{2\Omega_G} D . \tag{32}$$

This expression is valid for monochromatic operation while in the modelocking regime the field $f$ is a superposition of modes of different frequencies and then $D$ is time dependent in general. However $D$ can be split into its average value over a modulation period, $\overline{D}$, and a remainder, which we assume small. As this remainder has null average by definition its effect on the pulse velocity should be somehow neutral (clearly much less than that of $\overline{D}$) and affect mainly its shape. We thus approximate $\delta T_R$ by

$$\delta T_R \approx \frac{aw}{2\Omega_G} \overline{D} . \tag{33}$$

We note that an expression equivalent to (33) has been given by Haus[35].

**The field map**. We compute the field change

$$\delta f_{amp}(t) \equiv f(w, t + w/v) - f(0, t), \tag{34}$$

produced by the amplifier between an input field and its output (remind that $v$ denotes the group velocity of light in the amplifier), assuming that the change is small. Mathematically this is done through a Taylor expansion to the first order in the amplifier length $w$ which, in combination with the Maxwell Eq. (24a), yields

$$\delta f_{amp}(t) \approx (aw/2)p(0, t) + \delta T_R \partial_t f(0, t) . \tag{35}$$

Equating the right-hand sides of Eqs. (34) and (35) leads to an expression for $f(w, t + w/v)$ which can be plugged into the boundary condition (30). The result of this operation is the map

$$\begin{aligned} f(0, t + T_R) - f(0, t) = &-[\ell + m(t)]f(0, t) \\ &+ \ell r[1 - \ell - m(t)] \left[ p(0, t) + \frac{\overline{D}}{\Omega_G} \partial_t f(0, t) \right], \end{aligned} \tag{36}$$

where we introduced the usual laser pump parameter as

$$r \equiv aw/2\ell , \tag{37}$$

which is dimensionless and equals 1 at the free-running lasing threshold. Finally we approximate $\ell r[1 - \ell - m(t)]$ by $\ell r$ in the last term of Eq. (36) and arrive to the sought-for map given in Eq. (7) in the main text.

**The Gaussian pulses of Haus ME**. We look for a solution $F = F(\tau)$ and $\overline{G} = \overline{G}_0$ of Eq. (23) which is stationary with respect to the slow time $T$. For the electric field the solution is a Gaussian pulse in the fast time $\tau$[25]

$$F(\tau) = \sqrt{I_P} e^{i\phi_P} \exp\left[ -\frac{1}{2} \left( \frac{\tau - \tau_0}{\tau_P} \right)^2 \right] , \tag{38}$$

where $\phi_P$ is an arbitrary phase,

$$I_P = \frac{T_M}{\sqrt{\pi}\tau_P} \left( \frac{r}{\overline{G}_0} - 1 \right), \tag{39}$$

is the peak intensity,

$$\tau_P = \overline{G}_0^{1/4} / \sqrt{\mu \Omega_G \Omega_M}, \tag{40}$$

is the pulse duration, and

$$\tau_0 = -\tau_d \Omega_G / \left( 2\mu \Omega_M \overline{G}_0^{1/2} \right) \tag{41}$$

is the shift of the pulse peak from the lowest-loss point, proportional to the asynchrony $\tau_d$.

The lasing threshold gain $\overline{G}_0$ is solution of the equation

$$\overline{G}_0 - 1 - \mu(\Omega_M/\Omega_G)\overline{G}_0^{1/2} - \frac{(\Omega_G \tau_d)^2}{4\overline{G}_0} = 0, \tag{42}$$

and it is marginally greater than 1 in the realistic double limit $\Omega_M \ll \Omega_G \ll \tau_d^{-1}$.

Higher-order, Gauss-Hermite pulses exist as steady states of the problem[27], which however are always unstable[28]. The stability and dynamics of the Gaussian pulses have been studied, e.g., in refs. [28,66].

**The CME and the RNGH instability of free-running lasers**. Equations (20) with $\mu = \tau_d = 0$ (absence of loss modulation) govern the dynamics of a free-running laser. They admit the stationary solution $\{F, \overline{G}, g\} = \{F_s, \overline{G}_s, g_s\}$, with $|F_s|^2 = r - 1$, $\overline{G}_s = 1$, and $g_s = 0$, which represents the resonant singlemode lasing state. This solution exists for $r > 1$ and, as we show next, experiences the RNGH instability. For that, we perform a linear stability analysis by adding perturbations $\{\delta F(T, \tau), \delta G(T), \delta g(T, \tau)\}$, to the steady state $\{F_s, \overline{G}_s, g_s\}$, and keeping only terms linear in the perturbations. As the resulting Eqs. (20) (i) are linear, (ii) do not depend explicitly on $\tau$ (they are "translation invariant"), and (iii) verify periodic boundary conditions in $\tau$, the perturbations $\delta F(T, \tau)$ and $\delta g(T, \tau)$ can be written as Fourier series $\delta F(T, \tau) = \Sigma_\omega \delta F_\omega(T) \exp(-i\omega\tau)$, analogously for $\delta g(T, \tau)$ (remind that $\delta G(T)$ is not a function of $\tau$), so that only equal-frequency coefficients $\delta F_\omega(T)$, $\delta F_{-\omega}^*(T)$, and $\delta g_\omega(T)$ are (linearly) coupled. Note that the average $\overline{\exp(-i\omega\tau)} = 0$ because the perturbations must verify periodic boundary conditions in $\tau$ and we are considering $\omega \neq 0$ as we search for instabilities towards longitudinal modes different from the lasing mode.

Let us start with $\delta G(T)$, whose dynamical equation follows from Eq. (20b) and reads

$$\frac{d}{dT}\delta\overline{G} = -\frac{r}{T_G}\delta\overline{G}, \tag{43}$$

whose solution is $\delta\overline{G}(T) = \delta\overline{G}(0)\exp(\lambda_G T)$ with $\lambda_G = -r/T_G < 0$. Thus we set $\delta\overline{G} \mapsto 0$ in the following. Next, Eq. (20c) determines $\delta g_\omega(T)$ as

$$\delta g_\omega(T) = -\sqrt{r-1}\frac{1 + i\frac{\omega}{2\Omega_G}}{r - i\omega T_G}\left[\delta F_\omega(T) + \delta F_{-\omega}^*(T)\right]. \tag{44}$$

Note that Eq. (44), unlike (43), is an algebraic equation and not a dynamical equation, as discussed in the main text. Finally we consider the evolution of the field perturbations. From Eq. (20a) and using (44) we obtain

$$\frac{T_R}{\ell}\frac{\partial}{\partial T}\begin{pmatrix} \delta F_\omega \\ \delta F_{-\omega}^* \end{pmatrix} = -\begin{pmatrix} \frac{\omega^2}{\Omega_G^2} + c(\omega) & c(\omega) \\ c(\omega) & \frac{\omega^2}{\Omega_G^2} + c(\omega) \end{pmatrix}\begin{pmatrix} \delta F_\omega \\ \delta F_{-\omega}^* \end{pmatrix}, \tag{45}$$

where

$$c(\omega) = (r-1)\left(1 + i\frac{\omega}{\Omega_G} - \frac{\omega^2}{\Omega_G^2}\right)\frac{1 + i\frac{\omega}{2\Omega_G}}{r - i\omega T_G} . \tag{46}$$

It is easy to check that the perturbations' *amplitude* and *phase* quadratures, respectively defined as $\delta X_\omega = (\delta F_\omega + \delta F_{-\omega}^*)$ and $\delta Y_\omega = i(\delta F_\omega - \delta F_{-\omega}^*)$, evolve as $\delta Q_\omega(T) = \delta Q_\omega(0)\exp(\ell\lambda_Q T/T_R)$, $Q \in \{X, Y\}$. Here $\lambda_Y = -(\omega/\Omega_G)^2 < 0$, while $\lambda_X = -(\omega/\Omega_G)^2 - 2c(\omega)$ can become positive in a band of frequencies above a given pump threshold, signaling the growth of perturbations of appropriate frequency offset $\omega$, i.e., the RNGH instability. Considering the limit of interest $\gamma \equiv (\Omega_G T_G)^{-1/2} \ll 1$ (which defines so-called class B lasers), and normalizing the frequency offset as $\tilde{\omega} = (T_G/\Omega_G)^{1/2}\omega$ (motivated by the fact that the growing sidebands have frequencies of the order of $(\Omega_G/T_G)^{1/2}$[52–54,58]), we get

$$\text{Re}\lambda_X = \left[ 3(r-1) - \tilde{\omega}^2 - \frac{2r(r-1)}{\tilde{\omega}^2} \right]\gamma^2 + \mathcal{O}(\gamma^4). \tag{47}$$

For a pump $r > 9$, $\text{Re}\lambda_X > 0$ for $\tilde{\omega} \in [\tilde{\omega}_-, \tilde{\omega}_+]$ with

$$2\tilde{\omega}_\pm^2 = 3(r-1) \pm \sqrt{(r-1)(r-9)} , \tag{48}$$

exactly as in the RNGH instability of class B lasers[52–54,58,60].

To conclude we note that, on the contrary, Haus ME (23) with the same setting ($\mu = \tau_d = 0$) becomes a usual rate-equation description of laser dynamics which cannot account for the RNGH instability.

**Experimental methods**. The gain element of the ring semiconductor laser is a fiber-coupled traveling-wave high-power semiconductor optical amplifier (Superlum SOA-482) whose central wavelength is at 970 nm. It is operated in a polarization-maintaining fiber ring cavity which includes a 32 dB

polarization-maintaining optical isolator and a 10 GHz and 20 dB extinction-ratio lithium-niobate intensity modulator (Photline NIR-MX-LN-10), responsible for the loss modulation. The laser output is picked at a 10% reflectivity fiber beam splitter and guided to a 35 GHz photoreceiver (New Focus 1474-A). The substrate temperature of the gain medium is actively stabilized to 0.01 °C and all of the other components of the resonator are contained in a passive thermal stabilization enclosure.

The intensity modulator is operated around its linear region with an amplitude modulation close to 20 dB. At the lowest possible losses imposed by the modulator the laser threshold is at 23.5 mA and most of the measurements have been performed at a bias current of 30 mA. Additional tests performed at 36 mA did not reveal critical differences.

The modulation frequency was automatically scanned from 866.15 to 866.65 MHz by steps of 10 kHz. For each of these 50 values the photoreceiver signal has been acquired as series of ten million points by a single shot 33 GHz and 100 GS per second sampling rate oscilloscope (Tektronix DPO73304D). Another channel of the oscilloscope was used to monitor the loss modulation signal provided by a 12.5 GHz synthetizer (Rohde&Schwarz SMB-B112). The statistical and spatio-temporal analysis of the pulses have been realized on each single shot time trace using a numerical threshold crossing on the loss modulation signal as the origin of time for the pulse under consideration. In order to minimize the artificial jitter which would result from the discrete sampling of the oscilloscope, both the modulation and the detector traces have been linearly interpolated (in a mutually consistent way) such as to ensure zero jitter for the modulation signal while conserving fully and only the real jitter on the optical signal.

Around a modulation frequency of about 866 MHz each time trace contains about 85,000 periods corresponding to 25 optical pulses propagating over 3400 cavity round trips. To perform the statistical and spatio-temporal analysis shown in the next section we select one pulse out of 25 in the time series, which corresponds to actually analyzing always the same optical pulse one roundtrip after the other. Therefore the sample size for the statistical and spatio-temporal analysis is 3400, i.e., the number of roundtrips in one single-shot time trace. This analysis has been performed on several different optical pulses to check consistency.

**Numerical simulation methods**. We used two independent, standard algorithms to numerically simulate Eq. (20). One uses the split-step (Fourier) method while the other relies on a truncated modal expansion. Both methods give substantially identical results, confirming the validity of the assumptions. Here we illustrate them.

In the split-step method case it proves convenient to scale the slow and fast time variables $T$ and $\tau$ in Eq. (20) respectively to the cavity roundtrip time $T_R$ and to the time $\tau_{K-S} \equiv (\Omega_G \Omega_R)^{-1/2}$, which gives the order of magnitude of the pulse duration in the Kuizenga-Siegman theory; see (40). As Eq. (20c) is not a true dynamical equation, we can solve it at each time $T$, which we do perturbatively in the Fourier domain by neglecting at the lower order the nonlinear terms $g|F|^2$ and $\partial_\tau|F|^2$. The leading-order approximation to the fast gain obtained in that way is

$$\tilde{g}_0(T,\omega) = \overline{G}(T)\frac{\tilde{I}(T,0) - \tilde{I}(T,\omega)}{1 + i\rho\omega} , \tag{49}$$

where the tilde denotes the Fourier transform with respect to $\tau$, $\rho = (T_M T_G)/(\tau_{K-S} T_R)$, and $I(T,\tau) = |F(T,\tau)|^2$. The next order in the perturbative expansion is obtained by replacing $g$ with $g_0$ in the term $|F|^2 g$,

$$\tilde{g}_1(T,\omega) = \frac{\overline{G}(T)\tilde{\psi}(T,\omega) + \tilde{\phi}(T,0) - \tilde{\phi}(T,\omega)}{1 + i\rho\omega} , \tag{50}$$

where $\phi(T,\tau) = g_0(T,\tau)I(T,\tau)$ and $\psi(T,\tau) = \frac{1}{2}\partial_\tau|F(T,\tau)|^2$. By performing the inverse Fourier transform of Eqs. (49) and (50) we obtain $g_0(T,\tau)$ and $g_1(T,\tau)$ and we can finally write $g(T,\tau) = g_0(T,\tau) + g_1(T,\tau)$. We have checked numerically that the correction $g_1$ to the fast gain $g$ is indeed very small, which supports our perturbative treatment.

In this way we are left with Eq. (20a) for the electric field $F$ coupled to the mean-field Eq. (20b) for the slow gain $\overline{G}(T)$. The equation for the electric field can be integrated using the standard split-step Fourier method, where the linear part is solved in the frequency domain and the nonlinear one with an algorithm of the Runge-Kutta family. The same Runge-Kutta method is used to integrate the equation for $\overline{G}(T)$. Both equations depend on $g(T,\tau)$ which is computed at each instant $T$ of the slow time discretization as sketched above.

As for the truncated modal expansion, we introduce the new dimensionless slow time $t = T/T_R$, fast time $z = \tau/T_M + 1/2$ (which runs 0 to 1), and parameters

$$\eta = \frac{1}{\Omega_G T_M} \approx \frac{1}{\Omega_G T_R} , \Delta = \frac{\tau_d}{T_M} = \frac{T_M - T_R}{\ell T_M} ,$$
$$b = \frac{T_R}{T_G} , \sigma = \ell\frac{T_G}{T_R} = \frac{\ell}{b}. \tag{51}$$

Taking into account that $\Omega_M T_M = 2\pi$, Eqs. (20) become

$$\sigma^{-1}\partial_t F = \left[\overline{G} - 1 - 4\pi^2\mu^2(z - 1/2)^2 + \Delta\partial_z + \eta^2\partial_z^2\right]F \\ + gF - \eta\partial_z(gF) + \eta^2\partial_z^2(gF), \tag{52a}$$

$$\partial_t\overline{G} = r - \overline{G}\left(1 + \overline{|F|^2}\right) - \overline{g|F|^2} , \tag{52b}$$

$$b^{-1}\partial_z g = \overline{G}\left(\overline{|F|^2} - |F|^2\right) + \overline{g|F|^2} - g|F|^2 - g \\ + \eta\frac{\overline{G}}{2}\partial_z|F|^2. \tag{52c}$$

We now expand $F$ and $g$ in Fourier modes, owed to the periodic boundary conditions (16), i.e. $F(T, -T_M/2) = F(T, +T_M/2)$ and $g(T, -T_M/2) = g(T, +T_M/2)$:

$$F(t,z) = \sum_{n=-N}^{N} e^{i\alpha_n z}f_n(T) , \quad g(z) = \sum_{n=-N}^{N} e^{i\alpha_n z}g_n \tag{53}$$

with $\alpha_n = 2\pi n$, $g_n = g_{-n}^*$, and $g_0 = 0$ ($g(z)$ has no dc component). By projecting Eqs. (52a) and (52c) on the $n$-th mode we obtain

$$\sigma^{-1}\frac{df_n}{dt} = \left(\overline{G} - 1 + i\alpha_n\Delta - \eta^2\overline{G}\alpha_n^2\right)f_n - 4\pi^2\mu^2\int_0^1 e^{-i\alpha_n z}(z - 1/2)^2 F dz \\ + \int_0^1 e^{-i\alpha_n z}\left[gF - \eta\partial_z(gF) + \eta^2\partial_z^2(gF)\right]dz, \tag{54a}$$

$$\left(1 + i\frac{\alpha_n}{b}\right)g_n = -\int_0^1 e^{-i\alpha_n z}\left(\overline{G} + g\right)|F|^2 dz \\ + \frac{\eta\overline{G}}{2}\int_0^1 e^{-i\alpha_n z}\partial_z|F|^2 dz. \tag{54b}$$

Making use of modal expansions (53) and of integration by parts the above equations can be written as

$$\sigma^{-1}\frac{df_n}{dt} = \left(\overline{G} - 1 + i\alpha_n\Delta - \eta^2\overline{G}\alpha_n^2 - \frac{\mu^2\pi^2}{3}\right)f_n - 2\mu^2\sum_{m\neq n}\frac{f_m}{(n - m)^2} \\ + \left(1 - i\eta\alpha_n - \eta^2\alpha_n^2\right)\int_0^1 e^{-i\alpha_n z}gF dz, \tag{55a}$$

$$\left(1 + i\frac{\alpha_n}{b}\right)g_n + \sum_m\int_0^1 e^{i(\alpha_m - \alpha_n)z}|F|^2 dz \, g_m = -\left(1 - i\frac{\eta\alpha_n}{2}\right)\overline{G}\int_0^1 e^{-i\alpha_n z}|F|^2 dz. \tag{55b}$$

The last is a set of linear equations for the $g_n$'s that can be written in matrix form as $\hat{A}\mathbf{g} = \mathbf{F}$ , where the elements of the matrix $\hat{A}$ are

$$a_{m,n} = \left(1 + i\frac{\alpha_n}{b}\right)\delta_{m,n} + \int_0^1 e^{i(\alpha_m - \alpha_n)z}|F|^2 dz , \tag{56}$$

and the elements of the vector $\mathbf{F}$ are

$$f_n = -\left(1 - i\frac{\eta\alpha_n}{2}\right)\overline{G}\int_0^1 e^{-i\alpha_n z}|F|^2 dz . \tag{57}$$

We have verified numerically that the elements out of the main diagonal of $\hat{A}$ can be neglected so that we used the following approximation for the amplitudes $g_n$

$$g_n = -\frac{1 - i\frac{\eta\alpha_n}{2}}{1 + i\frac{\alpha_n}{b} + \sum_k|f_k|^2}\overline{G}\int_0^1 e^{-i\alpha_n z}|F|^2 dz . \tag{58}$$

## Data availability

The data that support the findings of this study are available from the corresponding author upon reasonable request.

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

## Acknowledgements

We thank E. Roldán and F. Mitschke for fruitful discussions. G.J.d.V., F.P., and A.M.P. acknowledge financial support from the Spanish Ministerio de Ciencia, Innovación y Universidades, and Agencia Estatal de Investigación, and the European Union FEDER (projects FIS2014-60715-P and FIS2017-89988-P). G.J.d.V. acknowledges financial support from the Generalitat Valenciana (grant BEST/2012/347). A.M.P. acknowledges financial support from the European Commission via a Marie Curie Fellowship (project ICONE-608099). Experiments were performed in the framework of the project OPTI-MAL granted by the European Union by means of the FEDER.

## Author contributions

G.J.d.V. conceived and set up the coherent master equation approach to laser mod-elocking. A.M.P., F.P. and G.J.d.V. derived the CME (20) and designed the numerical simulation methods. A.M.P. and F.P. performed the numerical simulations. B.G., F.G., and S.B. designed and performed the experiments. A.M.P., S.B., F.P. and G.J.d.V. discussed and analyzed the theoretical and experimental data. F.P. and G.J.d.V. wrote the manuscript in close collaboration with A.M.P. and S.B.

## Competing interests

The authors declare no competing interests.
