## [Peer Review File · Nature Communications]

Reviewers' comments:

Reviewer #1 (Remarks to the Author):

This is an interesting paper describing a new approach to describing mode-locked laser dynamics based on an extended "Master Equation" model that includes coherent atom-field coupling effects. The original Master Equation for modelocked lasers was developed by Haus, and constitutes a key physical model that describes pulse evolution in a mode-locked laser. The Haus Master Equation is one of the key foundations of dynamical laser theory, joining the quantic-cubic complex Ginzburg Landau equation as providing important analytical insights into laser operation that feed directly into the design of improved devices. Although mode-locked lasers are now a well-established technology, limits to pulse generation are continually being pushed, new diagnostics such as real-time measurements are being developed, and so the field continues to be of great interest.

A key message that I get when reading this paper is that whilst coherent effects in mode-locked lasers have been previously seen since the mid-1980s and 1990s in a range of systems (refs 26-34), the analysis that was used to interpret the results obtained was essentially a "first principles" approach based on the Maxwell-Bloch equations. Although able to reproduce experimental findings, such a theory is complex, and this complexity has hindered the appreciation of the importance of coherent effects in laser design, and potentially limited the development of new classes of laser system.

The major contribution of this paper is therefore to bring the complex phenomena associated with coherent transient effects (Rabi flopping etc) into the standard framework of the Haus master equation, both allowing new analytical insights to be obtained and providing a new and more efficient approach to numerical modelling.

Overall, this paper provides a greatly simplified and physical picture of results obtained in the past, and the authors also perform their own experiments to confirm their theory under controlled conditions in a different system. This work then provides significant new insight into mode-locked laser operation in a new operating regime, is of interest and I support its publication.

This said, I do wonder whether the authors have organized the paper in the best way for this paper to have optimal impact. Of course the decision on how they write the paper is up to them, but I think it could be more interesting to a wider readership if the discussion of the general field of coherent transients in optics (Allen & Eberly) and the historical review of results in Refs 26-34 came much earlier in the first few paragraphs, and then the "main result" in Equation 23 was simply stated with a physical discussion of the key parameters.

The present derivation is extremely clear from a nonlinear dynamics perspective, but it means that one gets to the potential impact of the work and the experiments much later. Maybe this is a question of personal taste, so I leave this for the authors (and editor) to consider – it would not require much work – simply shifting some of the theory into the Methods section, and adding a short discussion on how coherent effects manifest themselves physically.

It could also be appropriate to link this work to the current recent interest in dissipative soliton approaches to mode-locked lasers (theory and experiment).

I also think the paper may benefit by more closely relating this theory to the previous "precursor" observations of coherent effects. For example, their figure 5 is very similar to figure 2 of Ref 28. So can this CME theory now explain why the sech-profile was observed in these early experiments?

Also, the notion of Rabi oscillation has been proposed as a "power broadening" mechanism to increase the gain bandwidth in modelocked operation above the small-signal limit – if the CME

provides a new framework to design lasers that generate shorter pulses than expected by overcoming gain bandwidth limitations, then this should be highlighted much more – even in the abstract for example.

Reviewer #2 (Remarks to the Author):

This paper describes a new theoretical model for mode-locking of a laser that goes beyond the Ginzburg-Landau equation (the so-called master equation developed by Herman Haus and co-workers). Versions of the master equation that handle the cases of slow and fast effective saturable absorption exist, but they assume that the gain dynamics are slow on the time scales of the pulse evolution. The new model is able to handle fast gain dynamics, which occur most notably in semiconductor gain media. It will also handle completely coherent interaction of the light with the gain medium, but the focus in this paper is the treatment of media with fast gain dynamics. The authors adiabatically eliminate the polarization from the Maxwell-Boltzmann equations to generate what they call coherent master equations (CME) for the evolution. The reduction to one evolution equation offers some practical computational advantages and also illuminates deviations from the existing master equation. Examples are presented to show deviations from predictions of the master equation, and experiments on a fiber laser with a semiconductor gain medium agree well with predictions of the CME. The ratio of gain to round-trip lifetimes is varied by operating the laser on harmonics of the cavity repetition rate.

Gain media with short lifetimes can offer advantages in pulse-shaping, but also tend to foster instabilities in short-pulse generation. The CME could be applied to study these issues. There is significant interest in development of sources based on semiconductor gain media, including quantum-cascade lasers, so I think that there will be significant interest in this work. Dye lasers also have short gain lifetimes, but of course interest in them is minimal now. As the authors mention, a very interesting future extension would be to normal-dispersion systems with fast gain.

Overall, the work is motivated well, and the context is set appropriately. The paper is well-organized and clear. It is technical and specialized, but the overall points and conclusions are made well enough that a broad audience should be able to appreciate them. The experts will be able to dig in to the details.

Apart from a handful of typos and English errors that will be found by careful proof-reading, the paper could be published in its current form in my opinion. I have one comment on the presentation, which the authors may consider: The ability of the CME to handle the RGNH instability is emphasized in the abstract and discussion, and is even referred to as an “acid test” of the model. I suppose the comment refers to a qualitative phenomenon that is not predicted by the master equation. It seems strange that such an apparently-important point is only stated in one sentence in the body of the manuscript -- the argument is relegated to the Methods section.

Reviewer #3 (Remarks to the Author):

The authors present a new formulation of a so-called master equation that is used for the modeling of modelocked ultrafast lasers. Compared to the prior, very well established master equation approach by H. A. Haus, they managed to incorporate fast variations of the laser gain and light-matter coherence into this theoretical framework. The authors show how the traditional master equation emerges as a limiting case of their theory. Furthermore, they compare the predictions of Haus' master equation and the new formulation with experimental results from an actively modelocked semiconductor ring laser.

Haus' master equation has been tremendously successful in describing a wide range of modelocked lasers and covers most of the desirable operation modes of these lasers. However, a range of laser sources that are of present high interest and that promise to enable new industrial and scientific applications (mainly due to their compactness and the fact that they can in principle be mass produced) exhibit properties and behavior that may not be properly caught with Haus' approach. Specifically, this applies to modelocked semiconductor lasers, including the still controversially discussed quantum-cascade variety.

Judging the results from the perspective of all classes of modelocked lasers, it seems that the new formulation merely covers a niche of parameters and behaviors. The mentioned coherent and cooperative effects typically do not play a role for practical lasers. In view of the importance of the mentioned existing and emerging semiconductor sources, the presented result is clearly an important one and will have a significant impact in this field. Whether this is too specific to warrant publication in Nature Communications is an editorial decision. From the perspective of this specific field and the potential future relevance of the described lasers, the paper clearly deserves publication in a high-impact journal.

In the following, I will give my detailed comments:

a) Abstract, general remark: The new master equation can catch fast variations of the gain. Thinking of ultrabroadband modelocked lasers, fast variations may occur also in other cavity components and are typically not caught by Haus' master equation approach. To my understanding, this remains a limitation also of the new formulation (as long as parabolic approximations are used, the extendability towards ultrabroad bandwidths anyway remains limited). In general, a master equation approach assumes the changes in each element and on each roundtrip to be small, which ultimately doesn't hold anymore with very short pulses. Maybe the authors can comment on this?

b) First paragraph of introduction: maybe it is worthwhile to mention that some of these modelocking regimes/results are still controversial with respect to whether they constitute modelocking in the same strict sense as observed, e.g., in well-behaved solid-state lasers. The new master equation formulation could contribute to the resolution of this controversy.

c) Equation 5: The modelling of the gain profile assumes a homogeneously broadened laser line. This should be stated as it may not be applicable for other types of lasers.

d) Second paragraph of subsection titled "A new theoretical framework[.]": It might be worthwhile to state that the gain recovery time in solid-state and fiber lasers, which represent a large majority of today's ultrafast lasers, is typically on the order of microseconds to milliseconds. In these cases, the assumption of a static gain saturation is an excellent approximation.

e) Subsection II.A: It might be worthwhile to explicitly point out that the field in the master equation approach consists only of the slow envelope part of the pulse (i.e., after the fast carrier oscillation has been split off and removed).

f) Subsection II.A, description of gain: as typical for a master equation approach, the changes of the gain to the pulse are assumed to be small. The authors state that this follows from the necessary balance of gain and loss and that the losses are assumed to be small. This is an assumption that will not hold for high-power modelocked lasers, where, e.g., output coupling losses on the order of 20% or more are not uncommon. The authors should state this limitation as well.

g) Page 5, right column: The authors state that they obtained approximated expressions from Eq(16) with assumptions that they verified numerically. The authors should state for what

parameter ranges the validity of their approximations has been verified.

h) Fig. 2: The authors show the pulse profile obtained with their CME. The Haus Master equation for an actively modelocked laser with negligible dispersion and self-phase modulation predicts unchirped Gaussian pulses. To what extent does the "unchirped" property of the solution survive in the $T_G = 10$ ns and $T_G = 1$ ns case where Haus' master equation clearly fails?

i) Page 7, right column, second paragraph: The authors state that they use $T_G = 0.75$ ns for comparison with their experimental results. The authors should state where that number is coming from. Was this a kind of fitting parameter or is it obtained differently?

j) Page 7, right column, last two lines: The authors state that in the experiment the zero detuning condition is unknown. Couldn't that be measured with a radio-frequency spectrum analyzer?

k) Fig. 3 and Fig. 4: I'm surprised about the huge discrepancy in the time scale (horizontal axis) between model and experiment. The explanation that the authors give is not very satisfactory. They quote a paper from the early seventies ([47]) and state that the discrepancy might be due to "some inherent defect of every ME model". However, at least for picosecond actively modelocked solid-state lasers, Haus' master equation is known to predict pulse durations very accurately. I think, more effort is needed from the authors to explain this huge (nearly order-of-magnitude) discrepancy. After all, the authors introduce a new theoretical framework for modelling lasers. It should quantitatively deliver the right predictions in its validity range and its limitations should be clearly understood and documented in this paper.

l) Fig. 5: The experimental result seems to be well fitted by a sech-pulse. The authors argue that this is due to the master equation having a term that is similar to one that shows up in the master equation for passive modelocking via a fast saturable absorber. If this is the correct explanation, why does the theoretical results (Fig. 5a) match the sech pulse shape much worse? Could it be that the excellent match of the sech fit with the experimental data is just coincidental?

m) Section V. Discussion, first paragraph: it might be worthwhile to direct the reader once more to the methods section for the detailed discussion of the RNGH instability.

Coherent master equation for laser modelocking

Auro M. Perego, Bruno Garbin, Françoise Gustave,
Stephane Barland, Franco Prati, and Germán J. de Valcárcel

We thank the Reviewers for their valuable comments and for their overall appreciation of our work.

Before entering into details, we mention the changes done in our manuscript to meet the constraints imposed by the style of Nature Communications.

Structure of the manuscript:

We have divided the manuscript into the standard sections Introduction, Results, Discussion, Methods, and References.

We have eliminated the division in subsections everywhere, so that now only subheadings exist inside each part.

We have shortened the Introduction in order to respect the limit of 1,000 words and moved part of the material to Results (subsection “A. Gain dynamics and pulse broadening in ME approaches”). Also in the Introduction we have rephrased the last paragraph, which now begins with “In this paper...” and contains a summary of the main results that we present in our paper.

We have merged old Sections II, III and IV into Results.

We have rewritten to a large extent the Abstract in order to emphasize the novelty of the work and to highlight some of its expected applicability, motivated by the Reviewers’ comments.

References:

In order to respect the limit of 70 references (in the old version there were 75) we have removed 10 items because we have included 5 extra references in the new version. The removed items were numbered in the old version as [3], [4], [14], [39], [42], [63], [66], [71], [72], and [75]. The new references in the new version are numbered as [9], [26], [42], [63], and [68]. In particular [42] contains very preliminary results of this work appeared in a SPIE conference proceedings paper as far as in 2014, which now has been cited as indicated by the Editorial Policies. We didn’t include such a reference in our first submission because it contains very little information as compared to the present paper and actually we do not find it relevant here. We are attaching a copy of that conference paper for the Editor’s and Reviewers’ convenience.

Other changes:

Other changes that we made in response to the referees’ comments are explained in the detailed answers to them and are evidenced in blue in the revised paper. The changes evidenced in red were instead made just to improve the clarity of the presentation.

Reviewer 1 (Remarks to the Author):

We thank the referee for his/her appreciation of our work. In the second part of his/her report he/she suggests to change the organization of the paper: *This said, I do wonder whether the authors have organized the paper in the best way for this paper to have optimal impact. Of course the decision on how they*

write the paper is up to them, but I think it could be more interesting to a wider readership if the discussion of the general field of coherent transients in optics (Allen & Eberly) and the historical review of results in Refs 26-34 came much earlier in the first few paragraphs, and then the main result in Equation 23 was simply stated with a physical discussion of the key parameters.

The present derivation is extremely clear from a nonlinear dynamics perspective, but it means that one gets to the potential impact of the work and the experiments much later. Maybe this is a question of personal taste, so I leave this for the authors (and editor) to consider it would not require much work simply shifting some of the theory into the Methods section, and adding a short discussion on how coherent effects manifest themselves physically.

We thank the Reviewer for this suggestion, which we have partially accomplished as we explain next.

We have moved the historical discussion about coherent effects in laser mode-locking to the second paragraph of the Introduction, which we have enlarged by the inclusion of some sentences introducing the coherent transients in general (Allen & Eberly, new Ref. [9]): “...Coherent effects [9] occur when a stable phase relation is established between the light field and the electronic response of the material. Generally such a phase locking is fragile and coherent effects mostly manifest as transients. In laser systems, however, the long-term interaction between light and matter owed to the resonator feedback allows the persistence of such effects in some cases, e.g. through pulsed operation. We refer to ...”

We have moved to the Results section the more technical part about gain dynamics modelling and pulse broadening which was in the old Introduction.

Regarding the possible shift of some of the theory to Methods, we consider that the derivation of the model is indeed the core of our paper and we made an effort to make it as clear as possible, as the Reviewer him/herself recognises. We want to stress that we are in fact introducing a methodology which is an integral part of the new theory. Therefore we prefer to keep the derivation in the Results section (except the more technical part of the derivation of the map, which is somehow standard, which has been moved into Methods). In any case we do agree with the Reviewer that a kind of shortcut would be beneficial for some readers, and accordingly we have added a new paragraph just at the beginning of the Results section:

“In the first part of this section (up to Eq. (17) and its ensuing paragraph) we explain in detail the new formalism, which is then followed by the presentation of the final CME in Eq. (20), its analysis and comparison with the experiment. Hence, the reader more interested in the implications of the CME and its experimental verification can jump directly to Eq. (20), which is the CME for an actively modelocked two-level ring laser via amplitude modulation. For that reader’s convenience we note that the symbols ℓ , T_M , M , and Ω_G in Eq. (20) represent the resonator loss per roundtrip, the modulation period and depth, and the gain bandwidth, respectively, $\Omega_M = 2\pi/T_M$, while G and g are gain

components defined in Eqs. (18).”

Other changes suggested by the referee are the following:

It could also be appropriate to link this work to the current recent interest in dissipative soliton approaches to mode-locked lasers (theory and experiment).

In the final Section “Discussion” we have added a sentence and a reference to a paper where the dissipative soliton approach to modelocked lasers is presented “In particular, we observe that the very large ratio T_R/T_G needed for the emergence of dissipative solitons in the context of passive modelocking [68], should require using our CME properly adapted to the setup in order to reveal possible coherent effects.”

I also think the paper may benefit by more closely relating this theory to the previous precursor observations of coherent effects. For example, their figure 5 is very similar to figure 2 of Ref 28. So can this CME theory now explain why the sech-profile was observed in these early experiments?

A similar point about the sech-like profiles of the pulses was raised by Referee 3 (point 1). In response to this comment we have introduced two new parts in the text, in page 10.

In the first one (left column), following also a suggestion of Referee 3, we stress that the existence of sech-like pulses in active mode locking in our experiment should not be regarded as coincidental since it was observed previously, for instance in Ref. [12] (Ref. [28] in the old manuscript):

“Actually the experimental pulses are sech type, a fact already pointed out in early experimental studies of coherent effects of modelocking in argon lasers (compare our Fig. 5 with Fig. 2 of [12]). Our CME predicts pulses that interpolate between Gaussians and hyperbolic secants, and a clue of why this is happening comes from observing the main contribution to the pulse shape of the fast gain component g in Eq. (20a), which is $-\Omega_G^{-1}(\partial_\tau g)F$.”

The second comment (right column) is about the fact that in some cases in our CME the terms leading to seck-like pulses cannot be sufficiently large to counteract those leading to Gaussian pulses:

“Certainly there are additional terms in the CME (20) contributing to the pulse shape, notably the active modulation term proportional to τ^2 , which is responsible for the Gaussianity. Apparently in the experiment the role of the modulation on the pulse shape is less pronounced in certain cases, and this should be further considered in the future. In any case we can conclude that the CME (20) can explain, at least partially, why the sech profile was observed in early experiments [12], while our experimental observations indicate that such a qualitative change is robust.”

Also, the notion of Rabi oscillation has been proposed as a power broadening mechanism to increase the gain bandwidth in modelocked operation above the

small-signal limit if the CME provides a new framework to design lasers that generate shorter pulses than expected by overcoming gain bandwidth limitations, then this should be highlighted much more even in the abstract for example.

We thank the Referee for this comment. The fact that our CME can provide “a new framework to design lasers that generate shorter pulses” is now stressed at the end of the Abstract, that by the way was entirely rewritten, and in the last two sentences of the third paragraph of the Introduction:

“In other words, the rich and interesting phenomenology arising from coherent effects in lasers currently lacks a ME formalism, and this situation has probably hindered their deployment in laser design. A coherent ME theory would allow simple but rigorous description of laser operation in the presence of coherent effects, potentially paving the way to the development of new classes of laser systems that exploit light-matter coherence.”

(see also our answer to point a) of Referee 3)

Reviewer 2 (Remarks to the Author):

We thank the referee for his/her appreciation of our work. In the second part of his/her report he/she suggests the following changes:

Apart from a handful of typos and English errors that will be found by careful proof-reading, the paper could be published in its current form in my opinion.

We have carefully proof-read the paper and corrected some typos.

I have one comment on the presentation, which the authors may consider: The ability of the CME to handle the RNGH instability is emphasized in the abstract and discussion, and is even referred to as an acid test of the model. I suppose the comment refers to a qualitative phenomenon that is not predicted by the master equation. It seems strange that such an apparently-important point is only stated in one sentence in the body of the manuscript – the argument is relegated to the Methods section.

We thank the referee for this comment. Certainly, just one sentence in the main text about this result was at odds with its importance. As the Reviewer indicates the RNGH instability is a qualitative phenomenon that is not predicted by Haus master equation, which is important in some kinds of multimode and modelocked emission. Accordingly, the subsection devoted to the RNGH instability (page 6) has been enlarged giving enough details as to highlight its importance and as to make it readable without going to the Methods. In particular we have added a sentence in the first paragraph:

“This instability affects the singlemode lasing solution above a certain pump level, leading to spontaneous self modelocking, and is an acid test for coherent laser models.”

and a whole new paragraph at the end, where we summarize the result of the stability analysis:

“The CME (20) with $\mu = \tau_d = 0$ describes a free-running laser and accounts

for the usual singlemode lasing solution. A linear stability analysis of such a solution (see Methods) reveals that coupled sidebands, symmetrically detuned by $\pm\omega$ from the lasing mode, experience net gain above a threshold, signalling a multimode instability. The phenomenon is described by a complex eigenvalue of the linear problem, λ_X , whose real part governs the instability growth. In the relevant limit $\Omega_G T_G \gg 1$, it reads

$$\text{Re}\lambda_X = (\Omega_G T_G)^{-1} \left[3(r-1) - \tilde{\omega}^2 - \frac{2r(r-1)}{\tilde{\omega}^2} \right], \quad (22)$$

to the leading order, with $\tilde{\omega} \equiv \omega\sqrt{T_G/\Omega_G}$ the normalized sideband frequency. This expression matches the prediction of the full Maxwell-Bloch equations [52–55, 59] in the considered limit, which proves that the CME (20) contains the RNGH coherent instability. On the contrary, Haus ME doesn't account for this result.”

Reviewer 3 (Remarks to the Author):

We thank the Reviewer for his/her useful comments and appreciation of our work. He/she states that *the paper clearly deserves publication in a high-impact journal* but leaves to the Editor the decision about the appropriateness of Nature Communications as such a journal. We hope that the Editor finds this corrected version suitable for Nature Communications.

The Reviewer then lists a series of comments to which we provide next an answer.

a) Abstract, general remark: The new master equation can catch fast variations of the gain. Thinking of ultrabroadband modelocked lasers, fast variations may occur also in other cavity components and are typically not caught by Haus' master equation approach. To my understanding, this remains a limitation also of the new formulation (as long as parabolic approximations are used, the extendability towards ultrabroad bandwidths anyway remains limited). In general, a master equation approach assumes the changes in each element and on each roundtrip to be small, which ultimately doesn't hold anymore with very short pulses. Maybe the authors can comment on this?

Certainly our theory can catch fast variations of the gain. In principle this is not at odds with master equation approaches, whenever small losses are involved: fast variations need not be large variations from one roundtrip to the next one. In any case the fast variation question has to do mainly with how the polarization variable is adiabatically eliminated as the Reviewer points out. Here we use a parabolic expansion of the differential operator $\hat{\mathcal{L}}_t$ so as to keep the final master equation as simple as possible and make contact with Haus master equation in the proper limit. Yet, if necessary, higher order terms can be kept in the expansion or even the full Lorentzian operator can be used. We added two sentences about that in the paragraph after Eq. (9b):

”Expression (9b) represents the minimal expansion accounting for the finite

spectral bandwidth of the gain. In case of ultrashort pulses whose linewidth be comparable with the gain bandwidth one can extend the expansion or even keep the full differential operator as it is in Eq. (8). In particular, the latter would open the way to describe ultrashort pulses caused by coherent broadening [13].”

This question concerns the actual validity limits of our theory which, for sure, will have to be addressed in the future.

b) First paragraph of introduction: maybe it is worthwhile to mention that some of these modelocking regimes/results are still controversial with respect to whether they constitute modelocking in the same strict sense as observed, e.g., in well-behaved solid-state lasers. The new master equation formulation could contribute to the resolution of this controversy.

We thank the Reviewer for this useful comment. Actually, in the list of references in the first paragraph of the Introduction, which was mainly intended to show that the field of laser modelocking is still the subject of intense research, we put together references about modelocking and references about coherent multimode emission. To make clear that our new approach aims mainly at a better comprehension of coherent effects in laser modelocking and multimode emission we have moved immediately after the first paragraph of the Introduction the two paragraphs about coherent effects that were previously in page 2. We have also moved there Ref. [19] (previously Ref. [7]) which is focussed on coherent effects in multimode emission more than in modelocking.

c) Equation 5: The modelling of the gain profile assumes a homogeneously broadened laser line. This should be stated as it may not be applicable for other types of lasers.

The Reviewer is right and we have now stressed this limitation of our model in the first paragraph of Section II.B, observing by the way that the model can be easily generalized to include the effects of inhomogeneous broadening: ”We also note that inclusion of inhomogeneous broadening when necessary does not alter the skeleton of our approach.”

d) Second paragraph of subsection titled ”A new theoretical framework[.]”: It might be worthwhile to state that the gain recovery time in solid-state and fiber lasers, which represent a large majority of today’s ultrafast lasers, is typically on the order of microseconds to milliseconds. In these cases, the assumption of a static gain saturation is an excellent approximation.

The Reviewer is right. Yet we have found more convenient to add a comment on this in the second paragraph of Section II.I where we describe the results of the simulations with different values of the gain recovery time shown in Fig. 1: ”In Fig. 1 we show the results obtained for different values of T_G and fixed T_R . In the limit $T_G \gg T_R$, which is typical of solid-state, fiber or gas lasers (provided that the cavity is not extremely long) the gain saturation can be considered

constant across the pulse profile and the classic Gaussian pulse predicted by Haus ME is found.”

e) Subsection II.A: It might be worthwhile to explicitly point out that the field in the master equation approach consists only of the slow envelope part of the pulse (i.e., after the fast carrier oscillation has been split off and removed).

We thank the referee for this suggestion. We have modified accordingly the sentence after Eq. (7):

”Here $f(0, t)$ ($p(0, t)$) is the light field (electric polarization) complex slowly varying amplitude at the medium entrance plane ($z = 0$) . . .”

f) Subsection II.A, description of gain: as typical for a master equation approach, the changes of the gain to the pulse are assumed to be small. The authors state that this follows from the necessary balance of gain and loss and that the losses are assumed to be small. This is an assumption that will not hold for high-power modelocked lasers, where, e.g., output coupling losses on the order of 20% or more are not uncommon. The authors should state this limitation as well.

We agree with the referee. We have modified accordingly the second paragraph of Section II.C adding a comment in parentheses:

”Note that the assumption of a small field variation as caused by the amplifier is quite general (though it may not be satisfied in high power lasers with very large coupling losses): the amplifier action must just balance losses (or other variations) occurring outside the amplifying medium, and they are assumed small.”

g) Page 5, right column: The authors state that they obtained approximated expressions from Eq. (16) with assumptions that they verified numerically. The authors should state for what parameter ranges the validity of their approximations has been verified.

We thank the referee for this comment to which we have answered adding a new paragraph after Eqs. (21) in which we indicate the range of parameters for which we have assessed the validity of the approximations but we also stress that possible situations for which the approximations do not hold can be easily handled without changing the mathematical structure of the equations:

”We note that the validity of the final approximations (i) and (ii) above has been assessed numerically under diverse sets of parameters ($\Omega_G \in [10^{12}\text{s}^{-1}, 10^{13}\text{s}^{-1}]$, $T_G \in [0.5\text{ns}, 1\mu\text{s}]$, and $T_R \leq T_G$), which cover usual solid-state and semiconductor lasers. We observe that the approximations start to break down for much longer cavities, a situation that favors the size increase of the fast gain component g . In any case, if in a specific application approximations (i) and (ii) couldn’t be done, one would end up with a CME like (20) but with some more terms, which however would not change the mathematical structure of the

equations. Regarding such a structure, some comments are in order: ...”

Finally, the approximation about the modulation function, numbered as (2) in the old manuscript, has been moved upwards after Eq. (17), as that approximation is standard in ME approaches and has nothing to do with the other approximations, now numbered (i) and (ii).

h) Fig. 2: The authors show the pulse profile obtained with their CME. The Haus Master equation for an actively modelocked laser with negligible dispersion and self-phase modulation predicts unchirped Gaussian pulses. To what extent does the "unchirped" property of the solution survive in the $T_G = 10$ ns and $T_G = 1$ ns case where Haus' master equation clearly fails?

We would like to thank the Reviewer for the interesting comment. Like Haus' our theory admits real unchirped pulse solutions independently of the gain medium relaxation time. The CME can however be naturally generalized to describe the impact of group velocity dispersion and Kerr effect, as it is mentioned in the Section "Discussion". In those cases we expect to have situations where pulse solutions will be chirped.

i) Page 7, right column, second paragraph: The authors state that they use $T_G = 0.75$ ns for comparison with their experimental results. The authors should state where that number is coming from. Was this a kind of fitting parameter or is it obtained differently?

This value has been chosen as a reasonable guess but no attempts at measuring it have been made. Our numerical tests have shown no critical dependence on this specific value. For instance, we do not find any relevant difference for $T_G = 1$ ns and $T_G = 0.5$ ns.

j) Page 7, right column, last two lines: The authors state that in the experiment the zero detuning condition is unknown. Couldn't that be measured with a radio-frequency spectrum analyzer?

We thank the Reviewer for allowing us to improve the comment about this point because our previous wording was probably way too dry as we provided absolutely no reason for why it would not be possible to determine the zero detuning condition. Unfortunately the zero detuning condition is not easy to determine. Specifically, here we consider mode locking at a very high harmonic number (25) so that the spectra we observe are an incoherent superposition of 25 spectra, even in the locked state (since phases do not need to be identical for all sets of locked modes). The result is that the RF spectra provide only a very approximate value of the actual mode spacing, to the point that we could only indicate a zero detuning condition with uncertainties of tens of kHz, which is useless in a set of measurements covering few tens of kHz. We have added a brief discussion in the fifth paragraph of Section II.K:

“The detuning interval over which stable pulses can be found is similar in CME

simulations and in the experiment. In the experiment, outside the locked regime, the width of the peaks in the radiofrequency spectrum indicate that the mode spacing of the device is not defined to a better precision than a few tens of kHz. Therefore, the detuning condition can not be precisely determined, and the vertical axis in Fig. 3b is in absolute units.”

k) Fig. 3 and Fig. 4: I'm surprised about the huge discrepancy in the time scale (horizontal axis) between model and experiment. The explanation that the authors give is not very satisfactory. They quote a paper from the early seventies ([47]) and state that the discrepancy might be due to "some inherent defect of every ME model". However, at least for picosecond actively modelocked solid-state lasers, Haus' master equation is known to predict pulse durations very accurately. I think, more effort is needed from the authors to explain this huge (nearly order-of-magnitude) discrepancy. After all, the authors introduce a new theoretical framework for modelling lasers. It should quantitatively deliver the right predictions in its validity range and its limitations should be clearly understood and documented in this paper.

We thank very much the Reviewer for this comment about a weak point of our presentation. We have modified accordingly the paragraph of Sec. II.K that begins with “Such a discrepancy...”, adding a general reference to Siegman’s book and to his pioneering work with Kuizenga:

“Such a discrepancy was already observed in the earlier experiments on active modelocking, for instance in a Nd:YAG laser [26] and in a dye laser [37], and was ascribed to the etalon effect caused by the various intracavity elements. The theoretical treatment remains valid but the effective bandwidth of the atomic gain curve is drastically reduced and the pulsewidth is much wider than expected [63]. Specifically, we have determined that parasitic reflections at the amplitude modulator are responsible for the reduced spectral width. In addition, our model for a two-level system cannot fully capture the whole physics of a semiconductor amplifier, in particular by neglecting the linewidth enhancement factor (Henry’s α factor) of semiconductors, and this could have an impact on some important features of the pulses.”

l) Fig. 5: The experimental result seems to be well fitted by a sech-pulse. The authors argue that this is due to the master equation having a term that is similar to one that shows up in the master equation for passive modelocking via a fast saturable absorber. If this is the correct explanation, why does the theoretical results (Fig. 5a) match the sech pulse shape much worse? Could it be that the excellent match of the sech fit with the experimental data is just coincidental?

In response to this comment we have introduced two new parts in the text, in the last paragraph of Section II. In the first one, following also a suggestion of Referee 1, we stress that the existence of sech-like pulses in active mode locking in our experiment should not be regarded as coincidental since it was observed

previously, for instance in Ref. [12] (old Ref. [28]):

“Actually the experimental pulses are sech type, a fact already pointed out in early experimental studies of coherent effects of modelocking in argon lasers (compare our Fig. 5 with Fig. 2 of [12]). Our CME predicts pulses that interpolate between Gaussians and hyperbolic secants, and a clue of why this is happening comes from observing the main contribution to the pulse shape of the fast gain component g in Eq. (23a), which is $-\Omega_G^{-1}(\partial_\tau g)F$.”

The second comment is about the fact that in some cases in our CME the terms leading to sech-like pulses cannot be sufficiently large to counteract those leading to Gaussian pulses:

“Certainly there are additional terms in the CME (20) contributing to the pulse shape, notably the active modulation term proportional to τ^2 , which is responsible for the Gaussianity. Apparently in the experiment the role of the modulation on the pulse shape is less pronounced in certain cases, and this should be further considered in the future. In any case we can conclude that the CME (20) can explain, at least partially, why the sech profile was observed in early experiments [12], while our experimental observations indicate that such a qualitative change is robust.”

m) Section V. Discussion, first paragraph: it might be worthwhile to direct the reader once more to the methods section for the detailed discussion of the RNGH instability.

We thank the Reviewer for this comment, which was also a concern of Reviewer 2. Certainly the presentation of the RNGH instability in the previous submission was weak and highly dependent on the Methods section. In the new version of the paper we give much more details about the RNGH instability already in Section “Results” (see also our answer to Reviewer 2), so we think that now it’s no more necessary to refer to the Section “Methods” when we speak of the RNGH instability in the Discussion.

REVIEWERS' COMMENTS:

Reviewer #1 (Remarks to the Author):

I have read the revised manuscript as well as the response to the various reviews. Overall I think the authors have done a good job with addressing the various comments and from my perspective the paper can be accepted for publication. That said, there are still some places where the English could be improved from a stylistic perspective, but I leave this for the editorial team to look at and decide whether this is necessary.

Reviewer #2 (Remarks to the Author):

The revised manuscript is improved and it deserves to be published.

Reviewer #3 (Remarks to the Author):

The authors have considerably revised their manuscript. I'm fully satisfied with the response to my comments and the associated modifications of the manuscript. They have also carefully responded to the points raised by the other referees and revised the manuscript accordingly. I also agree with their additional changes to the manuscript (marked in red). The revisions have clearly led to an improved presentation.

With the text modifications, the authors have introduced one or the other typo. However, all of them are trivial and can easily be fixed during a potential production process of this paper.

Given that all my previous points have been fully addressed, I can only reiterate my previous recommendation that this work clearly deserves publication in a high-impact journal.